# Comparative analysis of bioinformatics tools to characterize SARS-CoV-2 subgenomic RNAs

Denise Lavezzari[1],*, Antonio Mori[1],*, Elena Pomari[1], Michela Deiana[1], Antonio Fadda[2], Luca Bertoli[2], Alessandro Sinigaglia[3], Silvia Riccetti[3], Luisa Barzon[3], Chiara Piubelli[1], Massimo Delledonne[2], Maria Rosaria Capobianchi[1],*, Concetta Castilletti[1],*

During the replication of severe acute respiratory syndrome coronavirus 2 (SARS-CoV-2), positive-sense genomic RNA and subgenomic RNAs (sgRNAs) are synthesized by a discontinuous process of transcription characterized by a template switch, regulated by transcription-regulating sequences (TRS). Although poorly known about makeup and dynamics of sgRNAs population and function of its constituents, next-generation sequencing approaches with the help of bioinformatics tools have made a significant contribution to expand the knowledge of sgRNAs in SARS-CoV-2. For this scope to date, Periscope, LeTRS, sgDI-tector, and CORONATATOR have been developed. However, limited number of studies are available to compare the performance of such tools. To this purpose, we compared Periscope, LeTRS, and sgDI-tector in the identification of canonical (c-) and noncanonical (nc-) sgRNA species in the data obtained with the Illumina ARTIC sequencing protocol applied to SARS-CoV-2–infected Caco-2 cells, sampled at different time points. The three software showed a high concordance rate in the identification and in the quantification of c-sgRNA, whereas more differences were observed in nc-sgRNA. Overall, LeTRS and sgDI-tector result to be adequate alternatives to Periscope to analyze Fastq data from sequencing platforms other than Nanopore.

## Introduction

The coronavirus infectious diseases-2019 worldwide pandemic caused by severe acute respiratory syndrome coronavirus 2 (SARS-CoV-2) is currently ongoing after being emerged from Wuhan (China) in December 2019 (Ecdc, 2020; Zhu et al, 2020; Spiteri et al, 2020). Up to now, SARS-CoV-2 has infected more than 750 million people and caused about seven million deaths worldwide (https://covid19.who.int/ last accessed July 05, 2023). SARS-CoV-2 is an enveloped positive-sense single-stranded RNA virus belonging to genus Betacoronavirus (order *Nidovirales*). During the replication of coronavirus genome, positive-sense genomic RNA (gRNA) and subgenomic RNAs (sgRNAs) are generated by the negative-sense RNA. The sgRNAs act as viral mRNA and are formed by a discontinuous process of transcription characterized by a template switch. A pivotal role in regulating the discontinuous transcription is played by transcription-regulating sequences (TRS) (La Monica et al, 1992; Hiscox et al, 1995; van Marle et al, 1995; Alonso et al, 2002; Thiel et al, 2003). The discontinuous transcription gives rise to a nested set of sgRNAs formed by 5′ UTR "leader" sequence fused to the "body" sequence derived from one of the 3′ structural or accessory genes. This transcribed sgRNA population is composed by canonical (c-sgRNAs) and noncanonical sgRNAs (nc-sgRNAs) (Ozdarendeli et al, 2001; Masters, 2006; Long et al, 2021). The synthesis of c-sgRNAs is based on TRS-L- and TRS-B-dependent discontinuous transcription, whereas nc-sgRNAs are a product of truncated fusions, frameshifted ORFs, and body-to-body junctions, (Kim et al, 2020; Long et al, 2021). Little is known about the makeup and dynamics of nc-sgRNA population and function of its constituents. Anyway, next-generation sequencing (NGS) approaches have made a significant contribution to expand the knowledge of sgRNAs in SARS-CoV-2 (Davidson et al, 2020; Doddapaneni et al, 2020 *Preprint*; Kim et al, 2020; Taiaroa et al, 2020 *Preprint*). In addition to NGS, other approaches have been used to elucidate, during infection in vitro or in clinical samples, their biology and the role as diagnostic and prognostic markers (Oranger et al, 2021; Stein et al, 2022; Telwatte et al, 2022). Currently, their diagnostic significance in clinical samples is not very clear (Wölfel et al, 2020).

As known, typical output of NGS consists of a "jumbo" quantity of data and requires powerful computational tools. In several studies, the SARS-CoV-2 transcriptome analysis was performed by using STAR (Spliced Transcripts Alignment to a Reference)-based approaches, that is, using RNA-seq aligner for the alignment of noncontiguous sequences directly to the reference genome (Dobin

[1]Department of Infectious and Tropical Diseases and Microbiology, IRCCS Sacro Cuore Don Calabria Hospital, Verona, Italy   [2]Department of Biotechnology, University of Verona, Verona, Italy   [3]Department of Molecular Medicine, University of Padova, Padova, Italy

Correspondence: elena.pomari@sacrocuore.it
*Denise Lavezzari, Antonio Mori, Maria Rosaria Capobianchi, and Concetta Castilletti contributed equally to this work

et al, 2013; Parker et al, 2021; Wang et al, 2021). However, STAR is not a specific bioinformatics tools for analyzing sgRNA, so suitable software have been specifically developed for detecting and quantifying sgRNAs, such as Periscope (Parker et al, 2021), sgDI-tector (di Gioacchino et al, 2022), LeTRS (Dong et al, 2022), and CORONATATOR (Lyu et al, 2022).

A brief description of the programs follows, **Periscope** (Parker et al, 2021) takes as input raw sequenced reads, without any preprocessing, derived from the ARTIC Nanopore-generated SARS-CoV-2 genome sequences, and was implemented to detect sgRNA from Illumina paired-end sequencing data produced by metagenomic, bait–capture-based or amplicon-based ARTIC Network. The software, based on a priori knowledge of TRS, recognizes reads that contain the SARS-CoV-2 leader using local alignment. **LeTRS** (Dong et al, 2022) identifies unique leader–TRS gene junction sites, analyzing Fastq files from either Illumina paired-end, or direct RNA/ARTIC Nanopore sequencing or BAM files, from any sequencing platform, produced by a spliced alignment method. LeTRS uses a Perl pipeline to identify the breaking site on quality-controlled and cleaned from adapters and low-quality reads Fastq files or on aligned BAM files. It identifies reads around a given known interval for leader-dependent c-sgRNAs and searches junctions out of the interval to identify reads with leader-independent fusion sites that suggest potential nc-sgRNA. Instead, **sgDI-tector** (di Gioacchino et al, 2022), differently from the other two software, uses an approach not based on previous knowledge of leader/junction TRS to analyze data derived from single-end sequencing from any sequencing platform. As matter of fact, the tool detects fragmented reads with insertion, deletion, copy-back, and hairping defective viral genome using the DI-tector (Beauclair et al, 2018),and picks out sgRNAs considering that sgRNAs coding for expressed ORFs are much more abundant than defective viral genome. Moreover, if the user provides an ORF reference, it can also identify sgRNAs considering the leader sequence and aligning them to the given reference to annotate the obtained data accordingly to user resources.

To stress, Periscope and LeTRS require previous knowledge of leader sequence and TRS, unlike sgDI-tector. All the three software can be adapted for viruses other than SARS-CoV-2.

Finally, **CORONATATOR** (Lyu et al, 2022) is a Perl and bash pipeline specifically implemented to quantify gene expression and identify in bona fide sgRNA in metatranscriptomic data from RNAseq-based coronavirus experiments. Summarily, its workflow consists of preprocessing aligned BAM files, breakpoint identification, sgRNA calling and profiling.

To the best of our knowledge, except for comparative evaluations performed by the developers of such software (di Gioacchino et al, 2022; Dong et al, 2022), no independent studies have been conducted to compare the performance of such bioinformatics tools. Thus, in the present study, Periscope (Parker et al, 2021), sgDI-tector (di Gioacchino et al, 2022), LeTRS (Dong et al, 2022) were tested to evaluate their performances in the identification of c- and nc-sgRNA using a dataset obtained with an amplicon ARTIC-based Illumina sequencing approach in SARS-CoV-2–infected Caco-2 cells. CORONATATOR was not tested as is more specifically designed for RNA-seq metatatrascriptomic data.

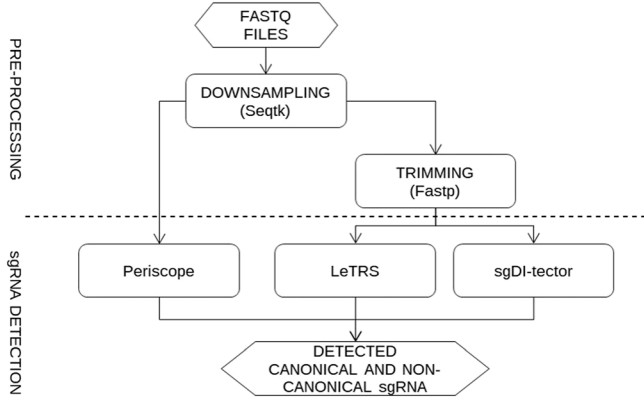

**Figure 1. The figure describes the procedure applied for the analysis of the data.**
The Fastq files were down-sampled to the same number of fragments and used as input for Periscope analysis, whereas were also trimmed before the analyses with LeTRS and sgDI-tector.

## Results

The NGS data from sequencing of infected Caco-2 cells were analyzed applying a pipeline (Fig 1) aimed at comparing the three software for the identification of sgRNAs in NGS data with a normalized number of fragments.

### Quality control of in vitro datasets

Caco-2 cells sequencing was conducted on five different time points after infection. All the samples presented a CT value > 10 (Fig 2A). Fig 2B (Table S1) shows the total number of sequenced fragments for each time point and for each replicate. All the samples exhibited a high number of fragments, ranging from ≃400,000 to ≃550,000 bp. The entire dataset showed high genome coverage, ranging from 6,096x to 7,034x. Fig 2C shows the coverage levels along the whole viral genome, with peaks in some regions. As displayed in Fig 2B (Table S1), all the samples had a high percentage of positions (>90%) covered by at least 1,000x reads, indicating that most of the positions were sufficiently covered.

### Canonical sgRNA concordance

To evaluate the concordance among the three software, each sample was down-sampled to the same number of fragments, that is, 421,872 initial fragments, as it was the lowest number of fragments in all our samples. Periscope, LeTRS, and sgDI-tector were inspected on the down-sampled data. As described elsewhere (Alexandersen et al, 2020; Kim et al, 2020), the number of fragments supporting the c-sgRNAs is a very small fraction of the number of initial fragments (Fig 3), ranging between 0.25% and 2.59% (Fig 4 and Table S2). In general, the two technical replicates for each time point agreed in the number of identified canonical sgRNAs and in the number of supporting fragments. Concordance among replicates have been evaluated calculating the Interclass Correlation Coefficient, data are shown in supplementary section, Table S2 (B and C). The highest percentage was observed in *ORFs, N>M>6>S*

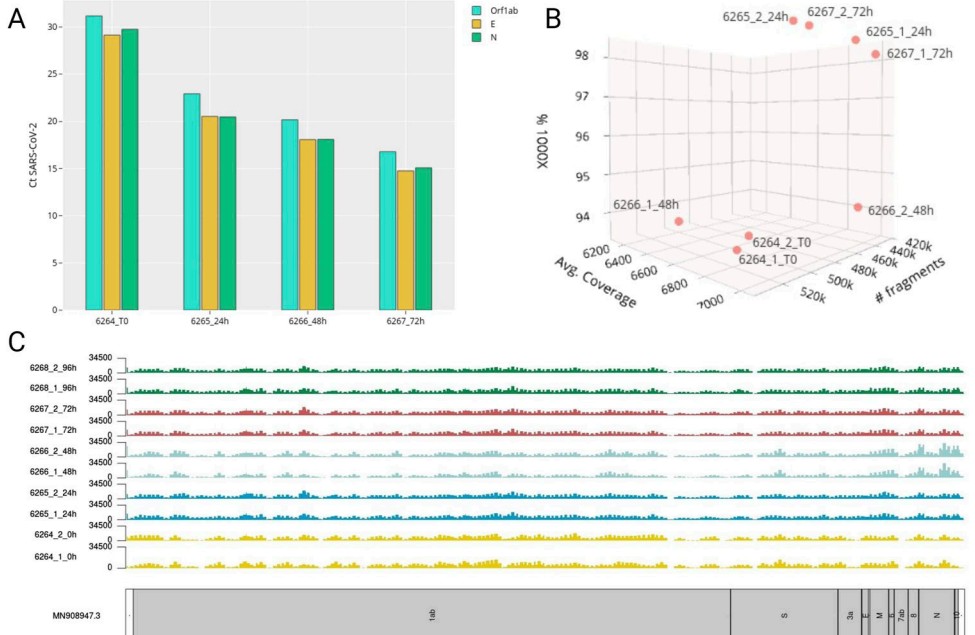

**Figure 2. Quality control results.**
**(A)** Caco-2 severe acute respiratory syndrome coronavirus 2 infection time course. For each sample (x-axis), the Ct values (y-axis) evaluated on the three genes (Orf1ab, E, N) is reported. **(B)** Correlation between the number of starting sequenced fragments (x-axis), average coverage (y-axis), and percentage of bases covered by at least 1,000 reads (z-axis) for each replicate and each time point. **(C)** Overall viral genome coverage for the two replicates at each time point. The two replicates are colored based on the postinfection time point: 0 hpi in yellow, 24 hpi in blue, 48 hpi in light blue, 72 hpi in red, and 96 hpi in green.

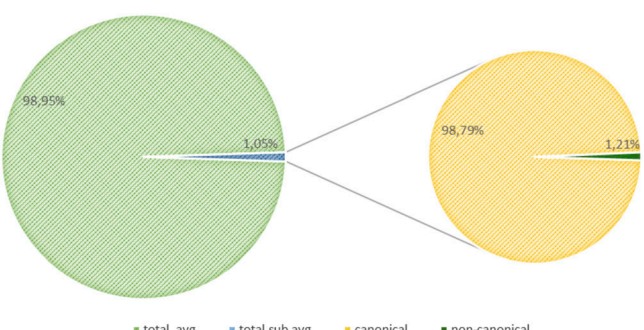

**Figure 3. Piechart summarizes overall (average of each timepoint and replicate) the average (avg) total number of genomic RNA fragments and the average total number of subgenomic RNA fragments.**
The pie chart on the right shows the subdivision of sgRNA fragments in c- and nc-sgRNA fragments.

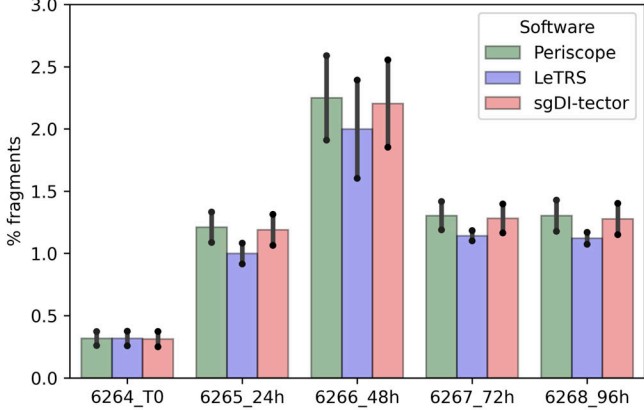

**Figure 4. Percentage of supporting counts (y-axis) for c-sgRNA for each software (Periscope in green, LeTRS in violet, sgDI-tector in pink) in each severe acute respiratory syndrome coronavirus 2-infected Caco-2 cell duplicate sample (x-axis) for each time point compared with the total number of fragments (3% is the highest percentage of fragments reached by sgRNAs).**
Bars represent the mean percentage between the two replicates at the same time point, whereas points show the single replicate value.

(Table S3). Notably, an increase in the number of fragments supporting these *ORFs* can be observed along the infection, peaking between 48 and 72 hpi (Fig S2I–K).

All three software were able to recognize the known c-sgRNA species present in all the infected cell samples. The concordance rate of identification between the software was calculated as the number of sgRNAs detected by all the three software divided by the total number of sgRNAs identified by the three software. As shown in Table 1, the three software showed a concordance rate close to 90% in most cases, except for the two replicates at 24 and one replicate at 96 hpi. In fact, these replicates exhibited the lowest concordance rate (77.78%).

The concordance was further investigated at single sgRNA species level along the infection course. Fig 5 shows, for each time point, the mean of the read counts between the two replicates

calculated with each of the three software. LeTRS, Periscope, and sgDI-tector exhibited high concordance in the sgRNA quantification for the principal ORF regions, that is, *ORFs S, M, 6, 7a,* and *N*. On the other hand, most of the differences between the three software were in the recognition of *ORFs 7b, 8,* and *10* sgRNAs. Single-sample progression is shown in Fig S1A–J.

The plots (Fig S2A–K) show the tendency of sgRNA distribution in ORF regions. The highest increment between time point T0 and 24 hpi was exhibit by *ORF7a*. *ORFs 6* and *S* show almost the same trend over time, with a slightly highest value at 24 hpi. *ORF N* shows a peak in the number of sgRNAs supporting fragments at 24 hpi.

**Table 1. Overall c-sgRNAs concordance rate between the three software for each replicate sample of severe acute respiratory syndrome coronavirus 2–infected Caco-2 cells.**

| Sample | Concordance rate (%) |
|---|---|
| 6264_1_T0 | 85.71 |
| 6264_2_T0 | 100.00 |
| 6265_1_24h | 77.78 |
| 6265_2_24h | 80.00 |
| 6266_1_48h | 87.50 |
| 6266_2_48h | 88.89 |
| 6267_1_72h | 87.50 |
| 6267_2_72h | 88.89 |
| 6268_1_96h | 88.89 |
| 6268_2_96h | 77.78 |

*ORF7b* was detected only by LeTRS and sgDI-Tector and shows the highest value at 24 hpi. *ORF E* showed a steady increase with the highest increment from 24 to 48 hpi. *ORF M* shows almost the same trend, with a slightly highest value at 48 hpi. *ORF3a* shows the highest number of sgRNA-supporting fragments at 48 and 96 hpi. sgRNAs for *ORFs 1ab*, *8* and *10* were virtually absent, being observed with only a few, if any, supporting reads at all time points (Fig 5A–E).

## Concordance of noncanonical sgRNAs

The presence of nc-sgRNAs was then investigated. The fraction of reads supporting nc-sgRNAs out of the total number of reads was between 0.001% and 0.027% (Fig 6 and Table S2), whereas out of the reads supporting total (canonical and noncanonical), sgRNAs calculated by the different software ranged between 0.81% and 5.08% (Table S4).

In the various experimental conditions analyzed here, only a small fraction of the total number of fragments, from 1 to 41 fragments, supported the detection of nc-sgRNAs (Table S5). The positions with the highest values were detected by all three software starting from samples collected at 24 hpi. In particular, in the *ORF 1ab* (*MN908947.3: 21,042-21,077*) and in the *ORF 7a/b* (*MN908947.3:27,744-27,779*) regions, LeTRS and sgDI-tector showed a relatively high number of supporting fragments. The highest

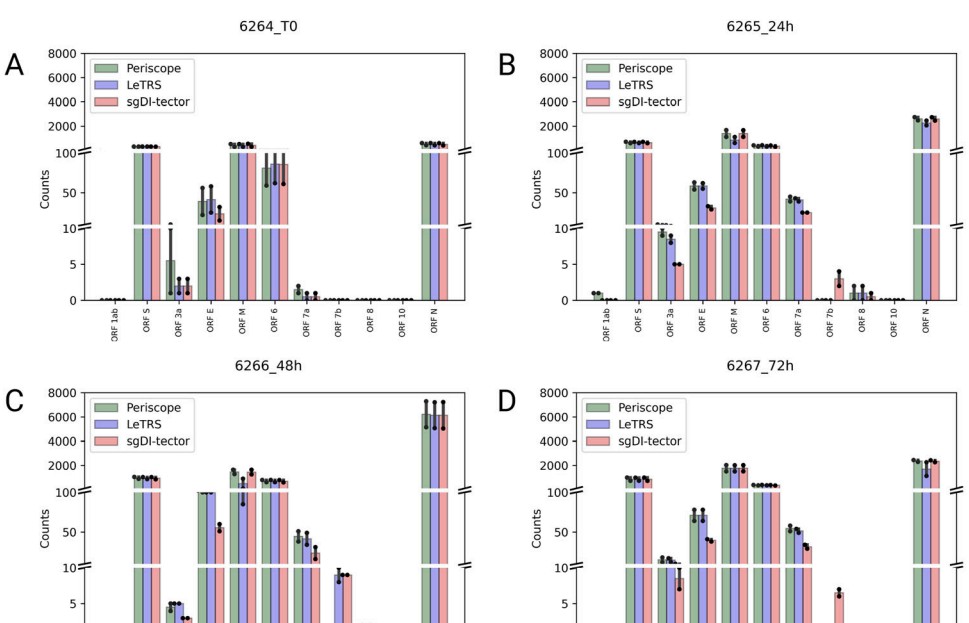

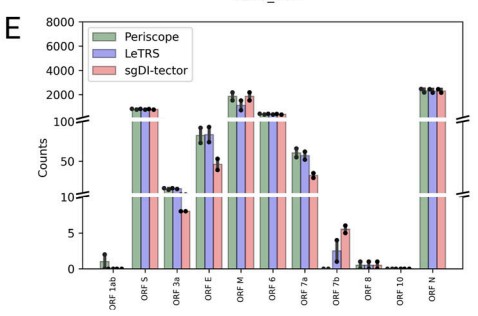

**Figure 5. Mean supporting c-sgRNAs counts at different time points.**
**(A, B, C, D, E)** Mean of supporting counts between replicates specified by each software (Periscope in green, LeTRS in blue, and sgDI-tector in red) for each indicated c-sgRNAs, in the samples of severe acute respiratory syndrome coronavirus 2–infected Caco-2 cells collected at the time points 0 hpi (A), 24 hpi (B), 48 hpi (C), 72 hpi (D), 96 hpi (E). Bars represent the mean counts between the two replicates at the same time point, whereas points show the single replicate value.

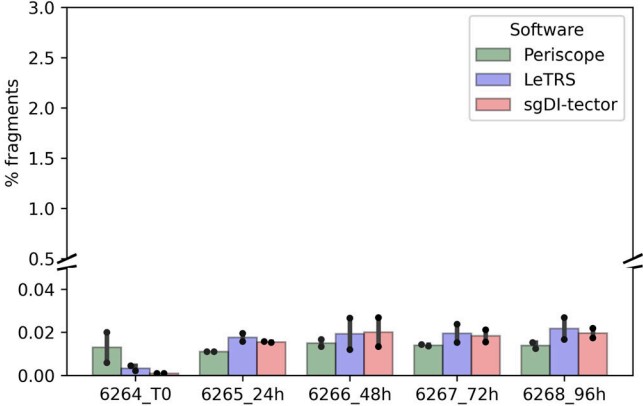

**Figure 6. Percentage of supporting counts (y-axis) for noncanonical sgRNAs for each software (Periscope in green, LeTRS in violet, sgDI-tector in pink) in each severe acute respiratory syndrome coronavirus 2–infected Caco-2 cell duplicate sample (x-axis) for each time point compared with the total number of fragments (3% is the highest percentage of fragments reached by sgRNAs).** Bars represent the mean percentage between the two replicates at the same time point, whereas points show the single replicate value.

**Table 2. nc-sgRNA concordance rate between of the three software for each replicate of severe acute respiratory syndrome coronavirus 2–infected Caco-2 cells.**

| Sample | Concordance rate (%) |
|---|---|
| 6264_1_T0 | 25.00 |
| 6264_2_T0 | 9.09 |
| 6265_1_24h | 47.06 |
| 6265_2_24h | 47.06 |
| 6266_1_48h | 46.34 |
| 6266_2_48h | 51.92 |
| 6267_1_72h | 51.06 |
| 6267_2_72h | 47.62 |
| 6268_1_96h | 53.06 |
| 6268_2_96h | 51.28 |

number (i.e., 41) of the supporting fragment was obtained in *ORF S*, position *MN908947.3: 22,271-22,291* only by Periscope at T0 that was collected at 1.5 hpi.

Also, for nc-sgRNAs, the concordance rate between the three software was calculated as the number of common nc-sgRNAs identified, divided by the total number of nc-sgRNAs identified by the three software. As shown in Table 2, in most of the samples, the concordance rate was around 50%, much lower than the one calculated for the c-sgRNAs, with the two replicates at T0 exhibiting the lowest concordance rate (9.09% and 25.00%, respectively).

The concordance between the three software was further investigated with Venn diagrams to evaluate the distribution of the identified nc-sgRNAs. Fig 7 shows that at T0, only few junctions were identified, with Periscope identifying the highest number of junctions. From time point 24 to 96 hpi, the three software presented a high number of identified junctions in common. In these cases, all

junctions identified by sgDI-tector were confirmed by at least one of the other two software, whereas LeTRS and Periscope showed a high number of nc-sgRNAs uniquely identified.

In addition, the ranges of junctions were examined in more details. To investigate the very small portion of nc-sgRNAs in the junctions' region, the scale of the below plots was set to a maximum of 100 fragments. Fig 8A–E shows the mean number of supporting fragments between the two time point replicates, the counts for each single replicate is reported as point in Fig 8A–E and as a bar chart in Fig S3A–J. As mentioned above, only few junctions were identified at T0, and most of them were detected by one software. From 24 hpi onward, a higher number of junctions were identified and most of them were consistent between the three software, with concordant numbers of supporting reads (Fig 8C–E). In particular, the region *MN908947.3: 27,744-27,779*, which contains the position 27,761 reported by Lyu et al (2022) to code for ORF7b, was confirmed in all the samples after 24 hpi, with high number of supporting reads and by all three software. Other two regions previously reported in the literature were confirmed by all three software. In particular, the region *MN908947.3:5,775-5,802*, containing the position 5,785 detected by Lyu et al (2022) and Parker et al (2021), was detected at 24 and 48 hpi and at 72 and 96 hpi with higher counts. The second region *MN908947.3:5,581-5,604*, containing the position 5,591 also identified by Lyu et al (2022), was present with only one or two supporting reads from samples at 48 hpi. Moreover, Periscope and sgDI-tector observed the regions: *MN908947.3:3,805-3,828* in at 48 hpi; *MN908947.3:14,651-14,681* in at 24 hpi and at 72 hpi; *MN908947.3:26,855-26,887* in from samples at 48 hpi, at 72 and 96 hpi. On the other hand, Periscope and LeTRS identified the region *MN908947.3:26,476-26,533* at 72 and 96 hpi, which contains the previously reported position *MN908947.3:26,868* (Lyu et al, 2022) (Fig S3).

Furthermore, all three software found different common regions that, as far as we known, were not reported in previous articles. In particular, the regions *MN908947.3:22,266-22,288* in *ORF S*, *MN908947.3:21,033-21,077* in *ORF 1ab* showed a high number of supporting reads starting from samples collected at 24 hpi.

Finally, the junction counts were grouped based on their genomic region and reported on the mean of the two time points (Fig 9A–E). Figs S4A–J and S5A–K report data for each replicate and on single ORFs. As shown in Fig 9A–E, in most cases the three software agreed on the number of supporting reads on the individual gene regions. In all samples, the nc-sgRNAs had a low expression in *ORFs E*, *M*, *N*, *6*, and *N*, whereas the corresponding c-sgRNAs showed a conspicuous presence (Figs 5A–E and S2A–J). At T0, only *ORFs 1ab*, *S*, *7a*, and *7b* were found with a low number of counts by all the software, except for *ORF S* which was detected only by Periscope (Fig 9A). Starting from 24 hpi, all replicates showed a meaningful nc-sgRNA expression in the genomic region *ORF S*, in line with the corresponding c-sgRNAs. Interestingly, all samples showed a significant expression of nc-sgRNAs in *ORFs 1ab*, *7a*, and *7b* that was opposite to the corresponding c-sgRNA levels. In addition, a low number of junctions were found in *ORF 10* from 24 hpi, whereas the c-sgRNAs from the same region were not detected at all (Fig 5A–E). Lastly, as for c-sgRNAs, *ORF 8* was found with a low number of supporting fragments from 24hpi.

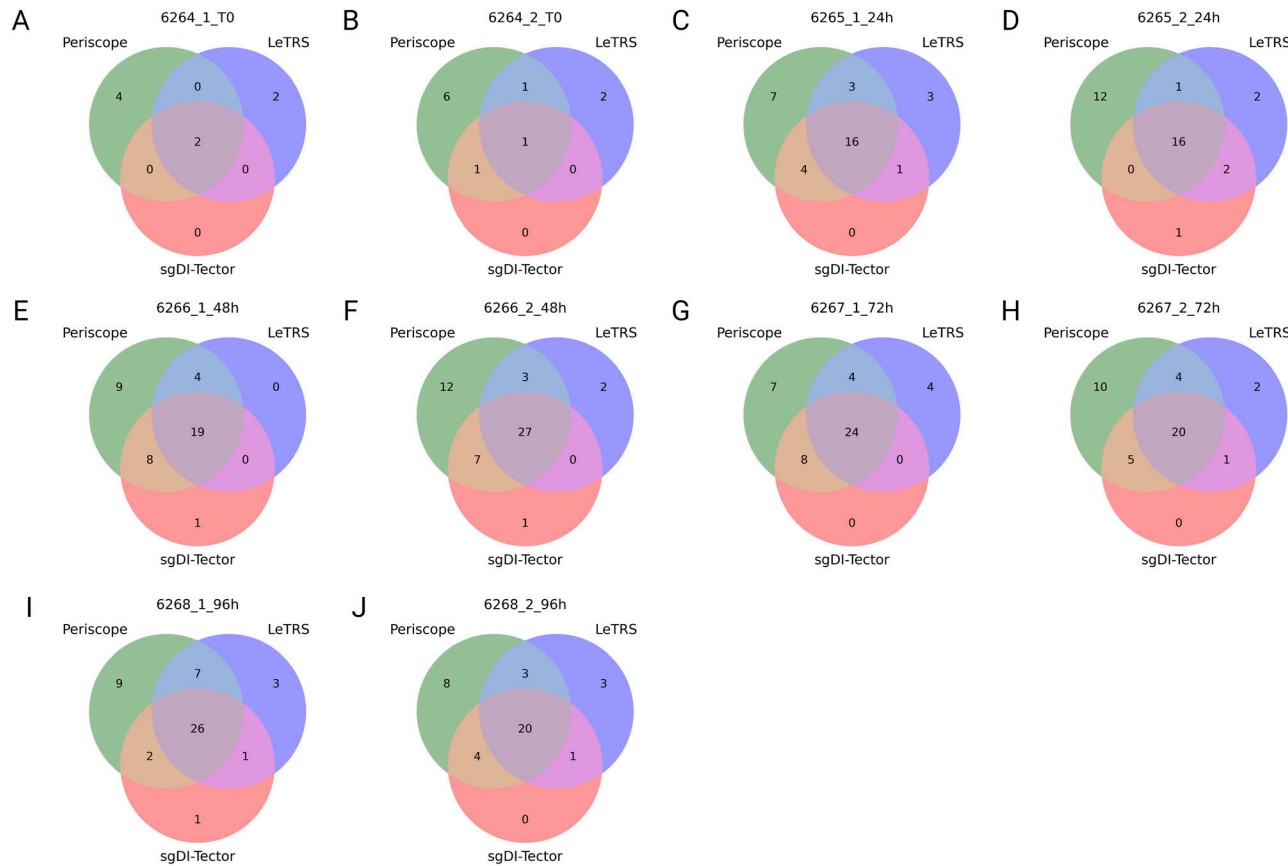

**Figure 7. Venn diagram of the nc-sgRNAs identified by the three software in replicate samples of severe acute respiratory syndrome coronavirus 2-infected Caco-2 cells collected at different time points.**
**(A, B, C, D, E, F, G, H, I, J)** (0 hpi replicate 1 (A), 0 hpi replicate 2 (B), 24 hpi replicate 1 (C), 24 hpi replicate 2 (D), 48 hpi replicate 1 (E), 48 hpi replicate 2 (F), 72 hpi replicate 1 (G), 72 hpi replicate 1 (H), 96 hpi replicate 1 (I), 96 hpi replicate 2 (J). Software are Periscope in green, LeTRS in blue, and sgDI-tector in red).

# Discussion

To better define SARS-CoV-2 trascriptome, several NGS studies (mostly, RNA-seq and amplicon-based) have been performed to investigate quantitative and qualitative variations of sgRNA population (i.e., c-sgRNAs and nc-sgRNAs). These studies are fundamental to deepen our knowledge about the contribution of sgRNAs in SARS-CoV-2 evolution, biology, and pathogenesis. Interestingly, amplicon-based sequencing is the main approach applied in SARS-CoV-2 epidemiological surveillance and genetic variant monitoring; consequently, its data are a treasure trove of information on the sgRNAs population in the various SARS-CoV-2 variants that have occurred during these 3 yr. Concerning bioinformatics tools, STAR (Dobin et al, 2013), Periscope (Parker et al, 2021), and CORONATATOR (Lyu et al, 2022) are tools not appropriate for the output from amplicon-based Illumina or Ion torrent sequencing; for this reason, more versatile software (such as LeTRS, sgDI-tector [Dong et al, 2022; di Gioacchino et al, 2022]) need to be applied. To recognize and profile c-sgRNAs and nc-sgRNAs, bioinformatics tools have to be precise and accurate to establish, especially regarding nc-sgRNAs, whose results represent true phenomena and not technical artefacts.

Until now, there are no studies that compare Periscope, LeTRS, and sgDI-tector on data generated by the ARTIC Illumina sequencing

approach. Here, we compared the results obtained with these three software, to assess their sensitivity when applied on sequencing data produced by a platform (Illumina) different from the Nanopore platform, with a particular interest on the sgDI-tector pipeline that can work without the requirement of a priori information. Specifically, we applied an amplicon ARTIC-based Illumina sequencing approach to analyze data derived from human epithelial colorectal cell line Caco-2 infected in vitro with SARS-CoV-2 lineage B.1, sampled daily until 96 hpi, and evidenced the changes in the proportion of various sgRNA species during the infection course. We used Caco-2 cells (i.e., human epithelial colorectal adenocarcinoma cell line) because they represent a more physiological system to study SARS-CoV-2 infection cycle, as opposed to the VERO cells (an African green Monkey kidney epithelial cell line), which do not produce alpha or beta interferons upon infection. Moreover, Caco-2 cells naturally express *ACE2* and *TMPRSS2* (Hoffmann et al, 2020), and SARS-CoV-2 infects and replicates significantly more efficiently than in the lung cell line Calu-3 with the same multiplicity of infection (Chu et al, 2020).

Regarding the raw sequenced data, the replicate outputs presented a high number of generated fragments. Analyzing the mapped short reads, all the samples presented a high depth (1,000x) covering 90% of the genome (Fig 2C).

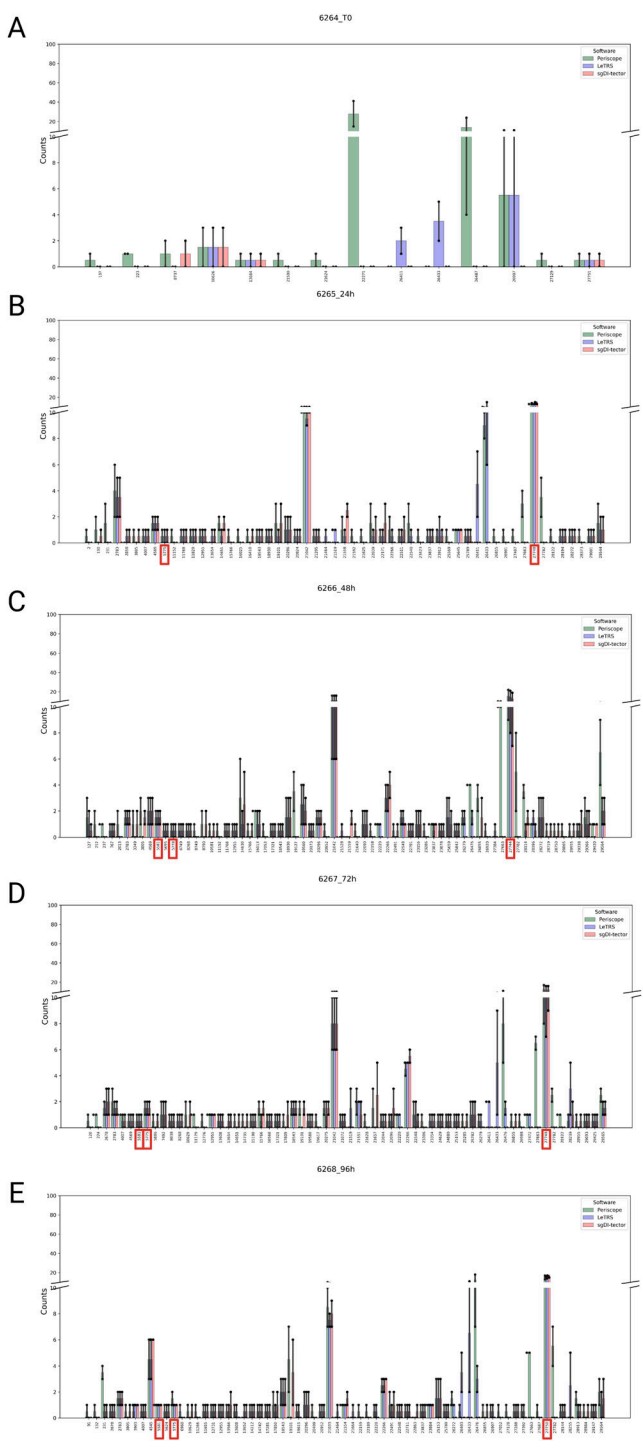

**Figure 8. Supported nc-sgRNAs counts specified by each software (Periscope in green, LeTRS in blue, and sgDI-tector in red).**
**(A, B, C, D, E)** Bars represent the mean counts between the two replicates at the same time point (0 hpi (A), 24 hpi (B), 48 hpi (C), 72 hpi (D), 96 hpi (E)), whereas points show the single replicate value. Red square represents positions already identified in the literature.

Concerning c-sgRNAs, our results showed a small percentage of fragments supporting sgRNAs with respect to the genomic reads, as previously described by Alexandersen et al (2020) and Kim et al

(2020). Nevertheless, the c-sgRNAs were consistently identified by the three software in the principal *ORF* regions at all the different time points (concordance rate ranging between 77.78% and 100%). Precisely, the three software exhibited a high concordance in the quantification of c-sgRNAs in the principal *ORF* regions (Figs 5A–E and S2A–K), that is, *ORFs N, M, 6, 7a*, and *S*, consistent with published articles which reports *ORF M* and *ORF N* as the most abundant c-sgRNAs produced during infection in vitro (Alexandersen et al, 2020; Bouhaddou et al, 2020; Doddapaneni et al, 2020 *Preprint*; Kim et al, 2020; Finkel et al, 2021). Moreover, *ORFs S, E, 3a, 6,* and *7a* were detected and quantified with similar values by all the three software in both replicates, with expression levels following all the explored temporal trends. Most of the differences among the three software were observed for *ORFs 7b* and *8* sgRNAs. These two sgRNAs do not rely on perfect homology of TRS short conserved core motif sequence, so their formation efficiency is considered low (Davidson et al, 2020; Kim et al, 2020; Parker et al, 2021; Lyu et al, 2022). *ORF 7b* c-sgRNA was observed only by LeTRS and sgDI-tector, possibly because both software are based on the direct detection of the breaking sites. *ORF 10* c-sgRNA was never observed by all three software, confirming previous reports (Davidson et al, 2020; Kim et al, 2020; Taiaroa et al, 2020 *Preprint*; Lyu et al, 2022), nevertheless, some literature data suggested the existence of ORF 10 (Finkel et al, 2021).

On the whole, the high agreement between programs shown in our comparison confirms that LeTRS and sgDI-tector are adequate software for c-sgRNA identification.

Regarding nc-sgRNAs, the total number of reads supporting them was only 5% (maximum) of the total reads supporting the sgRNAs (Table S4), consistent with a previous report (Lyu et al, 2022). The small percentage of reads is distributed along the different genomic *ORF* regions, with some peaks in the genomic *ORF region S* at T0, and in the four *ORF* regions *1ab, E, 7a,* and *7b* starting from 24 hpi (Fig 7). The similarity between the three software was analyzed considering a range of ±10 bp around the identified junction sites. The results showed a lower agreement between the three software (50% maximum) in comparison to that observed for c-sgRNAs. In particular, Periscope and LeTRS showed a high number of nc-sgRNAs individually identified. Investigating these differences, we noted that these junctions were supported by only few reads and, in general, these were not concordant between the two replicates, suggesting unreliable and unlikely nc-sgRNA identification, because of possible alignment or algorithm errors. On the other hand, three different regions identified by all three software (*MN908947.3:27,744-27,779, MN908947.3:5,775-5,802,* and *MN908947.3: 5,581-5.604*) contained junction positions previously reported in the literature (Parker et al, 2021; Lyu et al, 2022), supporting confidence on the results obtained in the present study (Fig 8). Moreover, observing the nc-sgRNAs with the highest quantification, the three software always agreed. Subsequently, the nc-sgRNAs were investigated based on the genomic *ORF* regions in which they occur (Figs 9 and S5A–K). The three software agreed for the detection and high quantification of nc-sgRNAs in genomic regions *ORFs 1ab, S, 7a,* and *7b* starting from 24 hpi. Moreover, all three software identified junctions in region *ORFs 3a, 8, 10,* and *N* with low supporting counts (Fig 9). The agreement of the three software in positions previously reported in the literature in the regions with high supporting counts

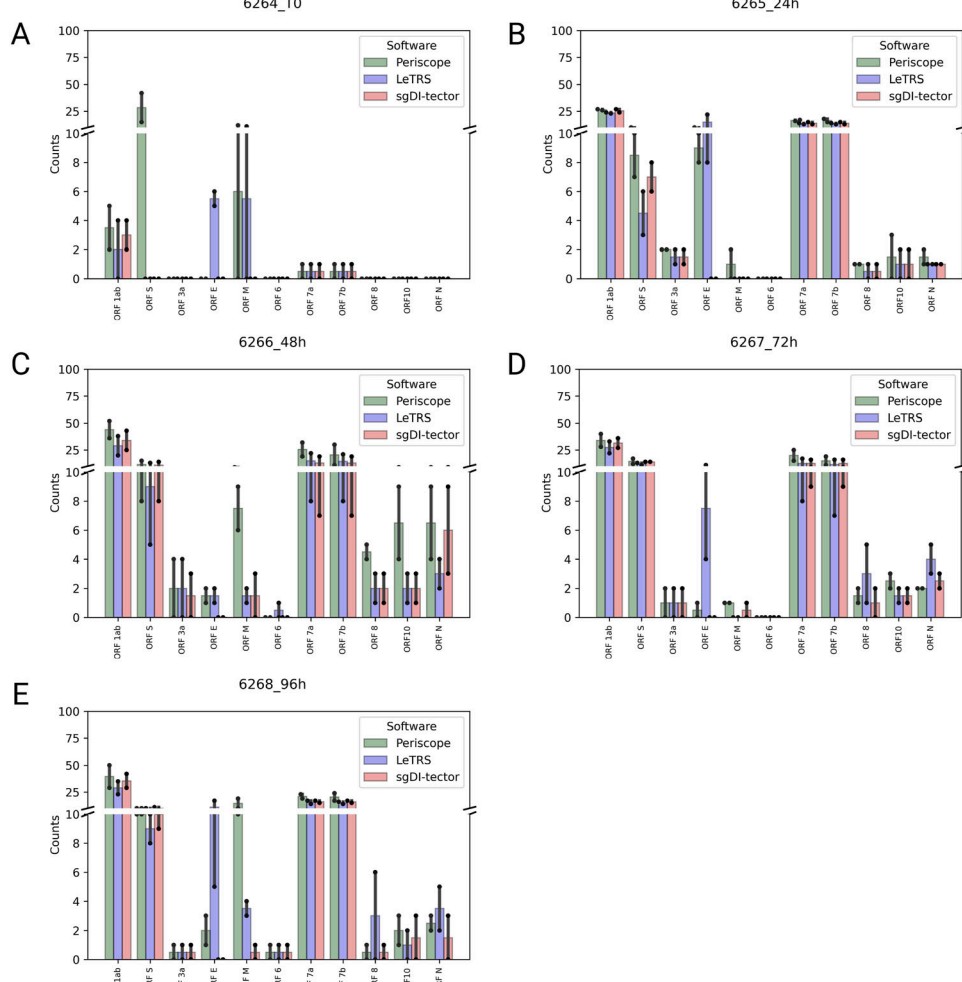

**Figure 9. Supported nc-sgRNAs counts for the indicated genomic *ORF* regions, specified by each software (Periscope in green, LeTRS in blue and sgDI-tector in red).** The plot is truncated to 100/421,872 as no data exceed that value. **(A, B, C, D, E)** Bars represent the mean counts between the two replicates at the same time point (0 hpi (A), 24 hpi (B), 48 hpi (C), 72 hpi (D), 96 hpi (E)), whereas points show the single replicate value.

and the genomic regions supports the idea that LeTRS and sgDI-tector are suitable tools also for the analysis of nc-sgRNAs.

Interestingly, the three software identified some different common nc-sgRNA regions never reported before. In particular, the regions *MN908947.3:22,266-22,288* in *ORF S*, *MN908947.3:21,033-21,077* in *ORF 1ab*, *MN908947.3:27,744-27,779* in *ORF 7a/b*, which were supported by a high-count number, suggesting that these putative nc-sgRNAs may be authentic features rather than artefacts.

The study presents limitations because there is no reference software available to confirm the presence or absence of the different sgRNAs, resulting in being impossible to assume through data as reference. To overcome this limitation, the fraction of sgRNA identified by each software was calculated posing at the denominator the total number of sgRNA species identified by any of the three software.

Even with this limitation, our findings from in vitro infected cells confirm LeTRS and sgDI-tector as good alternatives to Periscope for analyzing data from amplicon-based NGS sequencing methods with the aim of assessing amount and kinetics of sgRNA species in SARS-CoV-2–infected cells. Furthermore, two such software have the advantage to be applicable to different types of data. In

particular, sgDI-tector, that does not require a priori knowledge of TRS leader/junction, is suitable to identify nc-sgRNAs produced through a TRS-like homology-independent mechanism (Kim et al, 2020; Nomburg et al, 2020) reported three types of noncanonical junctions (TRS-L dependent, TRS-L independent distant, and local), all not originating from a known TRS-B. Hence, it is plausible that LeTRS and Periscope may not detect these junctions, especially the noncanonical TRS-L independent distant and local junctions.

In conclusion, we believe that it is advisable to conduct the sgRNA analysis using at least two software based on different analytical approaches in order not to lose useful information and to maximize the chances of identifying new putative nc-sgRNAs.

## Materials and Methods

### Cell culture and in vitro SARS-CoV-2 infection

Human epithelial colorectal adenocarcinoma cell line Caco-2 (Chu et al, 2020) (HTB-37; ATCC) was cultured in MEM (Thermo Fisher Scientific) supplemented with 20% FBS (Thermo Fisher Scientific),

1% penicillin–streptomycin (Pen/Strep; Thermo Fisher Scientific), and 1% GlutaMAX supplement (Thermo Fisher Scientific). For viral infection, the cells were seeded in a 24-well plate ($8 \times 10^4$ cells per well) 1 d before infection. Viral infection was performed at Biosafety Level 3 containment level. Briefly, before the infection, cell growth medium was removed, and cells were washed with DPBS (Thermo Fisher Scientific). Then, the virus, a SARS-COV-2 clinical isolate (GISAID number: EPI_ISL_15788618; Pango lineage B1), was added at a multiplicity of infection of 0.1 tissue culture infectious dose 50 (TCID50)/cell in MEM with 1% Pen/Strep. Cells were then incubated at 37°C with 5% $CO_2$ for 1.5 h to allow virus adsorption. Then, the cells were washed twice with 1× DPBS and incubated in DMEM (Thermo Fisher Scientific) supplemented with 2% FBS. Supernatants and cells were collected every 24 h thereafter, up to 96 hours postinfection (hpi) for RNA extraction and viral quantitation. All experiments were performed in three replicates. Uninfected Caco-2 cells were used as mock infection control.

### Virus titration

Supernatants of Caco-2 cells, collected at the various time points, were serially 10fold diluted up to $10^8$ in DMEM (Thermo Fisher Scientific) with 1% Pen/Strep and each dilution was inoculated in quadruplicate on Vero E6 cells which had been seeded ($2 \times 10^4$ cells/well) the day before in 96-well tissue culture plates. After 4–5 d, when the cytopathic effect was evident, the plates were stained with crystal violet fixing/staining solution (Sigma-Aldrich, Merck Millipore). The stained plates were extensively washed and then air dried. Presence/Absence of infecting virus was assessed by absence or presence of deep purple color in each well. The viral titer was calculated by the Spearman–Kärber algorithm, and expressed as 50% tissue culture infectious dose ($TCID_{50}$) (Spearman, 1908; Kärber, 1931).

### RNA extraction, library preparation, and NGS sequencing

RNA extraction from cell pellets was performed by MagnaNAPpure LC 2.0 System using the MagNAPure RNA isolation kit high performance (Roche), according to the manufacturers' instructions. To obtain a rough estimate of genomic viral RNA at each time point, RNA preparations (two replicates per condition) were merged into a single RNA pool per condition, then tested by RT–qPCR on a CFX96 Touch system (Bio-Rad Laboratories) with ANDiS FAST SARS-CoV-2 RT–qPCR Detection Kit (3D Biomedicine Science & Technology Co), targeting the protein gene regions N, E, and Orf1ab. RT–qPCR were used to estimate the genomic viral RNA content in each RNA pool on the basis of cycle threshold (Ct) values; samples showing a Ct value ≤ 30 were retrotranscribed and processed for NGS process.

Reverse transcription was performed as described elsewhere (Marcolungo et al, 2021). Two cDNA preparations were generated from each condition, and amplified using the ARTIC Primers v.1.3 (Marcolungo et al, 2021). PCR products were cleaned up using 1× AMPure XP beads (Beckman Coulter) and eluted in 15 $\mu$l of water. Amplicon libraries, prepared using the KAPA Hyper prep kit (Roche), were analyzed on the 4,150 TapeStation System (Agilent Technologies) and quantified using the Qubit dsDNA HS Assay kit (Thermo

Fisher Scientific). Libraries were sequenced on Illumina MiSeq in 250-bp paired-end.

### sgRNA detection

The Fastq files derived from infected cells underwent quality control using FastQC v.0.11.9 (Andrews, 2019). Data were downsampled at the same number of fragments using Seqtk v.1.3-r106 (https://github.com/lh3/seqtk, accessed on 28 November 2022). Adapters were trimmed using Fastp v.0.23 (Chen et al, 2018). Trimmed sequences were used as input for the sgRNAs analysis with LeTRS (options -t 4 -extractfasta -pool 0 -mode illumina -covcutoff 0 -primer_bed nCov-2019_primer.bed) (Dong et al, 2022) and sgDI-tector (di Gioacchino et al, 2022), whereas Periscope (–artic-primers V3 –resources periscope/resources_resource –technology illumina –threads 3) (Parker et al, 2021) was applied on raw downsampled sequenced data, as suggested by the authors. All three software were employed with default parameters. Because sgDI-tector only works with a single read, the tool was used separately for each read and the results were merged using a custom Python script (available at https://github.com/denise0593/sgRNA_comparison) that stored the highest count for each identified sgRNAs.

### Concordance rate calculation and performance assessment

Two Python scripts were developed to compare the different software results for c- and nc-sgRNAs. For the c-sgRNAs, the script compares the c-sgRNAs identified for each software, generates a Venn diagram, and calculates the concordance rate as the number of c-sgRNAs identified, divided by the total number of c-sgRNAs identified. In addition, the script produces a bar chart to analyze the number of reads supporting the identified c-sgRNA, that is, their abundance. For nc-sgRNAs, the developed script considers the junction point ±10 bp for all identified nc-sgRNAs, then merges the overlapping junctions using Bedtools v.2.27.1 and sums the supporting reads. These junction regions were compared between the three software and the concordance rate was calculated as the number of commonly identified regions divided by the total number of junction regions identified. The script produces the Venn diagram of the identified junctions and the bar graph of counts for each nc-sgRNAs. Moreover, the script groups the identified junction based on the belonging gene and creates a bar plot count in the corresponding ORF.

The interclass correlation coefficient was calculated using "irr" library v.0.84.1 in R v.4.3.1.

The scripts are available on GitHub (https://github.com/denise0593/sgRNA_comparison).

## Data Availability

The sequencing data generated in this study have been submitted to the NCBI BioProject database (https://www.ncbi.nlm.nih.gov/bioproject/) under accession number PRJNA934289.

# Supplementary Information

# Acknowledgements

This research was supported by EU funding within the MUR PNRR Extended Partnership Initiative on Emerging Infectious Diseases (Project No. PE00000007, INF-ACT), and by Italian Ministry of Health "Fondi Ricerca Corrente" to IRCCS Sacro Cuore Don Calabria Hospital. At the University of Padova, this work was supported in part by the European Union's Horizon 2020 research and innovation programme (VEO, Grant Number: 874735).

## Author Contributions

D Lavezzari: software, investigation, methodology, and writing—original draft, review, and editing.
A Mori: conceptualization, supervision, and writing—original draft, review, and editing.
E Pomari: conceptualization, supervision, investigation, and writing—review and editing.
M Deiana: investigation.
A Fadda: investigation.
L Bertoli: investigation.
A Sinigaglia: investigation.
S Riccetti: investigation.
L Barzon: investigation.
C Piubelli: supervision, methodology, and writing—review and editing.
M Delledonne: investigation.
MR Capobianchi: methodology and writing—review and editing.
C Castilletti: supervision, methodology, and writing—review and editing.

## Conflict of Interest Statement

The authors declare that they have no conflict of interest.

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
