## [Reviewer comments · Life Science Alliance]

Life Science Alliance

Comparative analysis of bioinformatics tools to characterize SARS-CoV-2 subgenomic RNAs

Denise Lavezzari, Antonio Mori, Michela Deiana, Antonio Fadda, Luca Bertoli, Alessandro Sinigaglia, Silvia Riccetti, Luisa Barzon, Chiara Piubelli, Massimo Delledonne, Maria Rosaria Capobianchi, Concetta Castilletti and Elena Pomari
DOI:<https://doi.org/10.26508/lsa.202302017>

Corresponding author(s): Dr. Elena Pomari (IRCCS Ospedale Sacro Cuore Don Calabria)

Review Timeline:

Submission Date:	2023-02-28
Editorial Decision:	2023-05-08
Revision Received:	2023-07-13
Editorial Decision:	2023-08-04
Revision Received:	2023-08-29
Editorial Decision:	2023-08-29
Revision Received:	2023-09-12
Accepted:	2023-09-13

Scientific Editor: Novella Guidi

Transaction Report:

May 8, 2023

Re: Life Science Alliance manuscript #LSA-2023-02017-T

Elena Pomari
IRCCS Ospedale Sacro Cuore Don Calabria
Viale L. Rizzardi 4
Negrar di Valpolicella 37024
Italy

Dear Dr. Pomari,

Thank you for submitting your manuscript entitled "Comparative analysis of bioinformatics tools to characterize SARS-CoV-2 subgenomic RNAs" to Life Science Alliance. The manuscript was assessed by expert reviewers, whose comments are appended to this letter. We invite you to submit a revised manuscript addressing the Reviewer comments.

Thank you for this interesting contribution to Life Science Alliance. We are looking forward to receiving your revised manuscript.

Sincerely,

B. MANUSCRIPT ORGANIZATION AND FORMATTING:

Reviewer #1 (Comments to the Authors (Required)):

Brief overview of the paper

Lavezzari et al. provide a comparison between three different tools aimed at the quantification of canonical (c-sgRNAs) and non-canonical sub-genomic RNAs (nc-sgRNAs) in SARS-CoV-2 infected CaCo2 cell lines. Beside the comparisons offered by the authors of the tools, a systematic benchmarking hasn't been published yet.

In this work, the different amounts of fragments assigned to each ORF annotated in literature is computed using the chosen softwares and concordance is evaluated both for c-sgRNAs and nc-sgRNAs. While c-sgRNAs quantification shows a good agreement throughout the compared tools, nc-sgRNAs quantification leads to more ambiguous results. This suggests using multiple softwares to investigate new putative junctions in the context of NGS data, since, being based on different principles, they maximize the information that can be obtained by this type of sequencing.

Although the work proposes an interesting point of view and a conceptually valid idea, it requires substantial revisions prior to publication. In particular the points listed below need to be addressed.

Major points

The work provides interesting insights into the performances of available tools for sgRNAs detection from NGS data. However, it may benefit from some improvements:

- Figures that are mainly made up of tables of fragment counts and percentages are not delivering a straightforward and quantitative message. I would like to suggest an easier data visualization approach, by using plots (e.g. concordance plots or barplots) such as in the case of Table 1, 2 and 4. These can be supplementary data, leaving the burden of delivering the message to specific plots that best address the scope.
- It would be helpful to provide a quantification control for the expression of different ORFs. In most of the plots, the expression results from the considered tools are compared by themselves but not to an orthogonal quantitative reference. Performing further experiments is not necessary, as data for sgRNAs quantification are already available in the form of Northern Blots (e.g. Ogando et al., doi: 10.1099/jgv.0.001453) or Nanopore Sequencing databases.
- CORONATOR tools has been named multiple times but never used in the comparison. It would be useful to have an explanation for this.
- Other tools mentioned in the LeTRS paper and available for the identification of TRS sites and junctions (such as SuPER and SARS-CoV-2-leader) were not included in the analysis. Which are the main differences between these tools and those that have been considered? Why were they excluded from the comparison?
- The provided github directory with the code is not accessible, therefore I could not review the adequacy and correctness of the analyses performed.

Minor points

Other more targeted suggestions may be the subsequent:

- Please, include citations for the introductory paragraph on SARS-CoV-2 (lines 39-45);
- Explain clearly why sgDI-detector "picks out all the sgRNAs considering that all sgRNAs include the leader sequence", but at the same time, its main principle is not to take into account any TRS sequence (lines 86-87);
- Table 1 might also include a general evaluation of the replicate concordance, including in the same measurement, both c-sgRNAs and nc-sgRNAs fragments;
- Correction at line 107 : it should be LeTRS and sgDI-detector;
- At lines 127-128, please explain the source of the total number of fragments chosen for normalization;
- Table 2 might be included in the supplementary material while a Figure expressing the percentage quantification for each replicate might substitute it, in order to deliver the message more straightforwardly.
- If the concordance of the two replicates at each time point is considered acceptable, from Table 3 onward, both replicates should be shown in the same plot. It might also be useful to explain the reason for having five different timepoints and what we would expect in terms of quantification of ORFs from each of them (see Major points).
- Include a plot for claim in lines 133-135 possibly with an orthogonal reference next to it.
- Please clarify, what do you mean in lines 145-146 for common sgRNAs. Does this refer to the independent sgRNAs to whom each ORF was assigned?

- It is not straightforward to extrapolate the meaning of Figure 4. It may benefit again from grouping the replicates for each tool and time point (and include single replicate information in the supplementary material. In addition, it would be nice to see as a supplementary figure, for each ORF, a barplot of the quantification across the five time points for the three tools, possibly next to an orthogonal reference (so to better represent claims of lines 158-162).
- Table 4 might benefit from including again, next to nc-sgRNAs quantification, data of c-sgRNAs quantification as percentages so to be able to compare all the information together through the same parameter.
- The take-home message of Figure 6 is not clear, and the interpretation may not be easy, especially because of low resolution images.
- Figure 7, as Figure 4, might benefit from the inclusion (as supplementary material) of barplots for each ORF across timepoints. Use a table to represent the quantity of new and previously annotated junctions in each sample at each time point, to simplify the representation.
- The discussion is well-articulated, even though, for each annotated junction, a close zoomed figure might help the reader to understand the data supporting the quantification performed by the three tools.

Reviewer #2 (Comments to the Authors (Required)):

In the manuscript, the authors compared the performance of three software tools (Periscope, LeTRS, sgDI-tector) for identifying subgenomic RNAs of SARS-CoV-2 using Illumina amplicon sequencing data sets obtained from SARS-CoV-2-infected Caco-2 cells. This is an interesting manuscript, but I got some suggestions and questions as the comments below.

1. I am glad to see that the authors have measured the Ct values of N, E, and Orf1ab in Table 1 for the test samples. This provides a good opportunity to determine the real copy number of N, E, and Orf1ab in these samples. I would suggest that the authors generate a standard curve using the Ct values to calculate the copy number of N, E, and Orf1ab, and then determine the ratio among them. This ratio can be compared to the ratio determined by bioinformatic tools.
2. It would be helpful if the authors could provide the primer sequences used in the RT-qPCR for N, E, and Orf1ab. This is crucial for reproducibility. Additionally, it would be important to know whether these primer pairs span the breaking site of subgenomic RNAs.
3. Table 3 shows that the three tools have a high concordance rate for counting subgenomic RNAs. However, I am curious whether these tools identified the same subgenomic RNAs from the same sequencing reads.
4. I understand that Periscope generates both high- and low-quality counts, while LeTRS identifies subgenomic reads with or without artic primers. It would be helpful if the authors could specify which outputs were used for the comparisons.
5. LeTRS can also identify leader-dependent novel subgenomic RNAs. I am curious whether these novel subgenomic RNAs were counted as non-canonical subgenomic RNAs in the analysis.

Reviewer #3 (Comments to the Authors (Required)):

The manuscript of D. Lavezzari and co-authors represents a comparative study of three bioinformatics tools, Periscope, LeTRS, and sgDI-tector, to detect, characterize, and quantify subgenomic RNAs (sgRNAs) in SARS-CoV-2-infected Caco-2 cells sampled at different time points. The authors highlight that although several bioinformatics tools have been developed to detect sgRNAs, limited studies have compared their performance. The authors have addressed this gap by comparing the performance of the three bioinformatics tools in identifying and quantifying canonical (c-sgRNA) and non-canonical (nc-sgRNA) subgenomic RNA species.

The authors report that the three software showed a high concordance rate in the identification and quantification of c-sgRNA, and consistent variations of individual c-sgRNA levels were observed along the infection course. Although the concordance rate was also high for nc-sgRNAs, some differences were observed compared to c-sgRNAs. Authors also conclude that LeTRS and sgDI-tector are adequate alternatives to Periscope to analyze Fastq data from sequencing platforms other than Nanopore.

The manuscript presents a well-structured study with clear research questions and methodology. The authors have used appropriate methods to analyze the data and have reported their findings comprehensively. However, the manuscript would benefit from a clearer description of methodology. In particular, it will be useful to know the settings and parameters that have been used to run the tools in the study. In addition, it is unclear why authors infected the cells at MOI=0.1, which would result in only 10% of the cells being infected. Although it is unlikely to impact the overall conclusion of the study, it seems more logical to me to infect cells at MOI=1 to ensure that authors are not dealing with a mixed population of cells at different stages of infection.

Dear Editor,

Thank you for considering our manuscript "Comparative analysis of bioinformatics tools to characterize SARS-CoV-2 subgenomic RNAs" by Lavezzari et al. for publication in *Life Science Alliance*. Please find enclosed the revised manuscript (using track changes), which has been modified according to the suggestions, and the 'Point-by-point response to reviewers' file. Figures are uploaded as separated files, tables are included at the bottom of the main manuscript.

We believe that, through the reviewers' suggestions, the new version of the paper has increased its scientific value for the readers of *Life Science Alliance*.

Sincerely yours,

Elena Pomari on behalf of all authors.

Dipartimento Malattie Infettive, Tropicali e Microbiologia

IRCCS Ospedale Sacro Cuore Don Calabria

Via Don A. Sempereboni, 5 - 37024 Negrar di Valpolicella (VR), Italy

tel. +39.045.601.3111 int 3095

elena.pomari@sacrocuore.it

Reviewer #1 (Comments to the Authors (Required)):

Brief overview of the paper Lavezzari et al. provide a comparison between three different tools aimed at the quantification of canonical (c-sgRNAs) and non-canonical sub-genomic RNAs (nc-sgRNAs) in SARS-CoV-2 infected CaCo2 cell lines. Beside the comparisons offered by the authors of the tools, a systematic benchmarking hasn't been published yet.

In this work, the different amounts of fragments assigned to each ORF annotated in literature is computed using the chosen software and concordance is evaluated both for c-sgRNAs and nc-sgRNAs. While c-sgRNAs quantification shows a good agreement throughout the compared tools, nc-sgRNAs quantification leads to more ambiguous results. This suggests using multiple software to investigate new putative junctions in the context of NGS data, since, being based on different principles, they maximize the information that can be obtained by this type of sequencing.

Although the work proposes an interesting point of view and a conceptually valid idea, it requires substantial revisions prior to publication. In particular the points listed below need to be addressed.

Major points

The work provides interesting insights into the performances of available tools for sgRNAs detection from NGS data. However, it may benefit from some improvements:

-Figures that are mainly made up of tables of fragment counts and percentages are not delivering a straightforward and quantitative message. I would like to suggest an easier data visualization approach, by using plots (e.g. concordance plots or barplots) such as in the case of Table 1, 2 and 4. These can be supplementary data, leaving the burden of delivering the message to specific plots that best address the scope.

Answer: We thank the reviewer for the comment. The tables 1, 2 and 4 were replaced as suggested with the Figures 2, 4 and 6 respectively. Tables were reported as Suppl. Table 1, Suppl. Table 2A and 2B respectively.

-It would be helpful to provide a quantification control for the expression of different ORFs. In most of the plots, the expression results from the considered tools are compared by themselves but not to an orthogonal quantitative reference. Performing further experiments is not necessary, as data for sgRNAs quantification are already available in the form of Northern Blots (e.g. Ogando et al., doi: 10.1099/jgv.0.001453) or Nanopore Sequencing databases.

Answer: We carefully read Ogando et al. paper but we are not sure that their transcripts relative abundance can be applied as orthogonal quantitative reference in our study. In fact, they used a different in vitro model and different time points (their last time point is 24hpi while in our study we considered a longer interval until 96hpi). Moreover, we would like to highlight that our NGS subgenomic analysis was performed through an amplicon approach that implies several PCR amplifications. For this reason, we believe that even if the genomic data had been available the correlation among the bioinformatic quantitation with the viral load calculated using a quantitative reference could be misrepresentative, that's why we would prefer to maintain a relative comparison between the three bioinformatics tools.

-CORONATATOR tools has been named multiple times but never used in the comparison. It would be useful to have an explanation for this. Other tools mentioned in the LeTRS paper and available for the identification of TRS sites and junctions (such as SuPER and SARS-CoV-2-leader) were not included in the analysis. Which are the main differences between these tools and those that have been considered? Why were they excluded from the comparison?

Answer: We thank the reviewer for the comment. We decided not to use SuPER and SARS-CoV-2-leader in our study since they were already compared with LeTRS (Xiaofeng Dong X. et al. Gigascience. 2022 May 26;11:giac045), showing lower sensitivity than LeTRS. Furthermore, SuPER and SARS-CoV-2-leader were mainly developed for RNA-Seq data. CORONATATOR was not tested as it is more specifically designed for analyzing RNAseq-based metatranscriptomic data. Finally, PERISCOPE has been used as the "gold standard" because it is widely used for SARS-CoV-2 subgenomic analyses, although it is not suitable for output from amplicon ARTIC-based Illumina sequencing approach. The main text was modified to further clarify this aspect (lines 123-124).

-The provided github directory with the code is not accessible, therefore I could not review the adequacy and correctness of the analyses performed.

Answer: We apologize for the inconvenience, but we are waiting the acceptance and publishing of the article to make the scripts public available.

Minor points

Other more targeted suggestions may be the subsequent:

-Please, include citations for the introductory paragraph on SARS-CoV-2 (lines 39-45);

Answer: We addressed this point, citations were added (line 55)

-Explain clearly why sgDI-detector "picks out all the sgRNAs considering that all sgRNAs include the leader sequence", but at the same time, its main principle is not to take into account any TRS sequence (lines 86-87);

Answer: As suggested we modified the description to clearly explain the pipeline (lines 105-109).

-Table 1 might also include a general evaluation of the replicate concordance, including in the same measurement, both c-sgRNAs and nc-sgRNAs fragments;

Answer: We thank the reviewer. The table 1 was replaced with the Fig.2 as suggested, and table 1 was reported as Suppl. Table 1. The number of fragments for c-sgRNAs and nc-sgRNAs was not reported in the Suppl. Table 1 as this table shows the quality of sequencing data.

-Correction at line 107: it should be LeTRS and sgDI-detector;

Answer: Done (line 468)

At lines 127-128, please explain the source of the total number of fragments chosen for normalization;

Answer: We added a sentence to better explain which source we used (lines 151-152).

-Table 2 might be included in the supplementary material while a Figure expressing the percentage quantification for each replicate might substitute it, in order to deliver the message more straightforwardly.

Answer: As previously specified we replaced as suggested with the Fig. 4, and table 2 was reported as Suppl. Table 2A.

-If the concordance of the two replicates at each time point is considered acceptable, from Table 3 onward, both replicates should be shown in the same plot. It might also be useful to explain the reason for having five different timepoints and what we would expect in terms of quantification of ORFs from each of them (see Major points).

Answer: We thank the reviewer for the comment. As suggested, we modified the Figure 5 and we kept the plots for each single replicate in the Suppl. Fig.1A-L and Suppl. Fig. 2A-M. As far as the different timepoints are taken into account, we know very well that from the point of view of the performance of the different tools, little changes. The reason why we decided to show all the timepoints is that we observed differences in expression over time and thought it could be appropriate to show these variations. About quantification, as already mentioned above, our NGS subgenomic analysis was performed by means of an amplicon approach involving several PCR amplifications; we think it is appropriate to keep only the relative comparison between the three bioinformatic tools.

-Include a plot for claim in lines 133-135 possibly with an orthogonal reference next to it.

Answer: We thank the reviewer for the suggestion. We included a new plot (Fig.5 and Suppl. Fig.1A-L and Suppl.Fig. 2A-M). As discussed in a previous comment, we prefer not to use an external quantitative reference based on literature data.

-Please clarify, what do you mean in lines 145-146 for common sgRNAs. Does this refer to the independent sgRNAs to whom each ORF was assigned?

Answer: We added a sentence to better clarify the concept (line 166)

-It is not straightforward to extrapolate the meaning of Figure 4. It may benefit again from grouping the replicates for each tool and time point (and include single replicate information in the supplementary material). In addition, it would be nice to see as a supplementary figure, for each ORF, a barplot of the quantification across the five time points for the three tools, possibly next to an orthogonal reference (so to better represent claims of lines 158-162).

Answer: As suggested, we modified the Figure 4 now Fig. 5 and we kept the plots for each single replicate in the Suppl. Fig.1A-L. Moreover, we added in the Suppl.Fig. 2A-M the plot for each ORF across the five time points. As discussed in a previous comment, it was not possible to use an external quantitative reference based on literature data.

-Table 4 might benefit from including again, next to nc-sgRNAs quantification, data of c-sgRNAs quantification as percentages so to be able to compare all the information together through the same parameter.

Answer: We thank the reviewer for the comment. The table 4 was replaced as suggested with the Fig. 6 and was reported as Suppl. Table 2B.

-The take-home message of Figure 6 is not clear, and the interpretation may not be easy, especially because of low resolution images.

Answer: See comment below.

-Figure 7, as Figure 4, might benefit from the inclusion (as supplementary material) of barplots for each ORF across timepoints. Use a table to represent the quantity of new and previously annotated junctions in each sample at each time point, to simplify the representation.

Answer: As suggested we modified the Figure 6 now Fig. 8 and we kept the plots for each single replicate for each position in the Suppl. Fig. 3A-L and grouped by ORFs in the Suppl. Fig. 4A-L. Moreover, we added in the Suppl. Fig. 5 A-M the plot for each ORF across the five time points.

The quantity is reported in the Suppl. Table 3B.

-The discussion is well-articulated, even though, for each annotated junction, a close zoomed figure might help the reader to understand the data supporting the quantification performed by the three tools.

Answer: Thanks to the previous suggestions, we added more punctual figures and the discussion was modified consequently.

Reviewer #2 (Comments to the Authors (Required)):

In the manuscript, the authors compared the performance of three software tools (Periscope, LeTRS, sgDI-tector) for identifying subgenomic RNAs of SARS-CoV-2 using Illumina amplicon sequencing data sets obtained from SARS-CoV-2-infected Caco-2 cells. This is an interesting manuscript, but I got some suggestions and questions as the comments below.

1. I am glad to see that the authors have measured the Ct values of N, E, and Orf1ab in Table 1 for the test samples. This provides a good opportunity to determine the real copy number of N, E, and Orf1ab in these samples. I would suggest that the authors generate a standard curve using the Ct values to calculate the copy number of N, E, and Orf1ab, and then determine the ratio among them. This ratio can be compared to the ratio determined by bioinformatic tools.

Answer: We thank the reviewer for the comment. While we find the suggestion intriguing, it should be noted that RT-qPCR was used to evaluate the genomic viral RNA content. As such, we established a cut-off of Ct value ≤ 30 to determine eligibility for NGS analysis; these specifics are outlined in the Methods section 4.3 of our manuscript. As suggested, we performed a standard curve for N gene for which we have a quantified standard and, we used the curve to calculate the viral load of each time point (please refer to the graph and the table reported below). Unfortunately, it was not possible for us to calculate the viral load using Orf1ab and E because we do not have a standard for those genes, therefore, we cannot determine the ratio among the three genes. In terms of NGS data we would like to highlight that our bioinformatic analysis focused solely to investigate subgenomic regions, hence, we cannot calculate the bioinformatic ratio among N, E and Orf1ab genes, neither compare the viral load determined with the standard curve with this data. Furthermore, the subgenomics analysis was performed through an amplicon approach which implies several PCR amplifications. For this reason, we believe that even if the genomic data had been available, the correlation among the bioinformatic quantitation with the viral load calculated using the standard curve (only N gene) could be misrepresentative.

VIRAL COPIES	CQ
2*10 ¹	36.18
2*10 ²	31.51
2*10 ³	28.80
2*10 ⁴	23.91
2*10 ⁵	20.50
2*10 ⁶	16.90
2*10 ⁷	13.00
2*10 ⁸	9.50

TIME POINT	VIRAL COPIES	CQ N GENE
T0	754	29.76
24H	220297	20.48
48H	935355	18.1
72H	582331	15.09
96H	9018724	14.37

2. It would be helpful if the authors could provide the primer sequences used in the RT-qPCR for N, E, and Orf1ab. This is crucial for reproducibility. Additionally, it would be important to know whether these primer pairs span the breaking site of subgenomic RNAs.

Answer: Thanks to the referee for the advice. RT-qPCR were performed using ANDiS FAST SARS-CoV-2 RT-qPCR Detection Kit (3DMed Diagnostics) to quantify the genomic viral RNA extracted. As this is a commercial kit, the primer sequence is confidential and not available, so we are unable to determine where the primers map.

3. Table 3 shows that the three tools have a high concordance rate for counting subgenomic RNAs. However, I am curious whether these tools identified the same subgenomic RNAs from the same sequencing reads.

Answer: We thank the reviewer for the comment. Unfortunately, this information cannot be obtained because LeTRS and sgDI-tector outputs do not give information about unique matching information between reads and sgRNA (i.e., which reads is responsible for which sgRNA), hence this comparison is not feasible.

4. I understand that Periscope generates both high- and low-quality counts, while LeTRS identifies subgenomic reads with or without artic primers. It would be helpful if the authors could specify which outputs were used for the comparisons.

Answer: We thank the reviewer for the comment. PERISCOPE generates both high- and low-quality counts but only when Nanopore data are used as input; in our case, as we used Illumina data, only one output was available. For LeTRS we used the option with ARTIC primers; we specified it in the methods (line 475).

5. LeTRS can also identify leader-dependent novel subgenomic RNAs. I am curious whether these novel subgenomic RNAs were counted as non-canonical subgenomic RNAs in the analysis.

Answer: We thank the reviewer for the comment. The answer is affirmative; the leader-dependent novel sgRNAs were considered non-canonical on the conditions that originating from unknown TRS-B.

Reviewer #3 (Comments to the Authors (Required)):

The manuscript of D. Lavezzari and co-authors represents a comparative study of three bioinformatics tools, Periscope, LeTRS, and sgDI-tector, to detect, characterize, and quantify subgenomic RNAs (sgRNAs) in SARS-CoV-2-infected Caco-2 cells sampled at different time points. The authors highlight that although several bioinformatics tools have been developed to detect sgRNAs, limited studies have compared their performance. The authors have addressed this gap by comparing the performance of the three bioinformatics tools in identifying and quantifying canonical (c-sgRNA) and non-canonical (nc-sgRNA) subgenomic RNA species.

The authors report that the three software showed a high concordance rate in the identification and quantification of c-sgRNA, and consistent variations of individual c-sgRNA levels were observed along the infection course. Although the concordance rate was also high for nc-sgRNAs, some differences were observed compared to c-sgRNAs. Authors also conclude that LeTRS and sgDI-tector are adequate alternatives to Periscope to analyze Fastq data from sequencing platforms other than Nanopore.

The manuscript presents a well-structured study with clear research questions and methodology. The authors have used appropriate methods to analyze the data and have reported their findings comprehensively. However, the manuscript would benefit from a clearer description of methodology. In particular, it will be useful to know the settings and parameters that have been used to run the tools in the study. In addition, it is unclear why authors infected the cells at MOI=0.1, which would result in only 10% of the cells being infected. Although it is unlikely to impact the overall conclusion of the study, it seems more logical to me to infect cells at MOI=1 to ensure that authors are not dealing with a mixed population of cells at different stages of infection.

Answer: We thank the reviewer for the suggestion; we added the parameters in the methods (lines 475-477). We totally agree with the reviewer that a MOI allowing a single cycle might have been the best choice. At the time of experimental design, we thought to choose the 0.1 MOI to perform a long infection kinetic.

August 4, 2023

Re: Life Science Alliance manuscript #LSA-2023-02017-TR

Dr. Elena Pomari
IRCCS Ospedale Sacro Cuore Don Calabria
Viale L. Rizzardi 4
Negrar di Valpolicella 37024
Italy

Dear Dr. Pomari,

Thank you for submitting your revised manuscript entitled "Comparative analysis of bioinformatics tools to characterize SARS-CoV-2 subgenomic RNAs" to Life Science Alliance. The manuscript has been seen by the original reviewers whose comments are appended below. While the reviewers continue to be overall positive about the work in terms of its suitability for Life Science Alliance, some important issues remain.

Our general policy is that papers are considered through only one revision cycle; however, given that the suggested changes are relatively minor, we are open to one additional short round of revision. Please note that I will expect to make a final decision without additional reviewer input upon re-submission.

Please submit the final revision within one month, along with a letter that includes a point by point response to the remaining reviewer comments.

To upload the revised version of your manuscript, please log in to your account: <https://lsa.msubmit.net/cgi-bin/main.plex>
You will be guided to complete the submission of your revised manuscript and to fill in all necessary information.

B. MANUSCRIPT ORGANIZATION AND FORMATTING:

Sincerely,

Reviewer #1 (Comments to the Authors (Required)):

The work presented by Lavezzari et al. has improved from the first revision, however there are still some points that may benefit from clarifications and expansions. Finally, I would suggest a professional proof reading.

Major points

-Figures that are mainly made up of tables of fragment counts and percentages are not delivering a straightforward and quantitative message. I would like to suggest an easier data visualization approach, by using plots (e.g. concordance plots or barplots) such as in the case of Table 1, 2 and 4. These can be supplementary data, leaving the burden of delivering the message to specific plots that best address the scope.

Answer: We thank the reviewer for the comment. The tables 1, 2 and 4 were replaced as suggested with the Figures 2, 4 and 6 respectively. Tables were reported as Suppl. Table 1, Suppl. Table 2A and 2B respectively.

Reviewer's answer: the inclusion of the new figures delivers a much clearer message to the reader, thank you for implementing them. Furthermore, in every plot where data is represented by a mean of two duplicates, the authors should show, beside the mean, also the points representing the value of each replicate (e.g. <https://doi.org/10.1038/s41551-017-0079>). A mean of two points is qualitatively delivering an information but it is not considerable statistically robust.

-It would be helpful to provide a quantification control for the expression of different ORFs. In most of the plots, the expression results from the considered tools are compared by themselves but not to an orthogonal quantitative reference. Performing further experiments is not necessary, as data for sgRNAs quantification are already available in the form of Northern Blots (e.g. Ogando et al., doi: 10.1099/jgv.0.001453) or Nanopore Sequencing databases.

Answer: We carefully read Ogando et al. paper but we are not sure that their transcripts relative abundance can be applied as orthogonal quantitative reference in our study. In fact, they used a different in vitro model and different time points (their last time point is 24hpi while in our study we considered a longer interval until 96hpi). Moreover, we would like to highlight that our NGS subgenomic analysis was performed through an amplicon approach that implies several PCR amplifications. For this reason, we believe that even if the genomic data had been available the correlation among the bioinformatic quantitation with the viral load calculated using a quantitative reference could be misrepresentative, that's why we would prefer to maintain a relative comparison between the three bioinformatics tools.

Reviewer's answer: Given that the PCR creates itself an artefact on the quantification, it is exactly for this reason that I would like to see an orthogonal method (a single timepoint at 24 hpi is sufficient, and other authors have quantified sgRNAs in CaCo2, as <https://doi.org/10.1016/j.molcel.2021.02.036>). Indeed, this will suggest how close or how far is the quantification of the three proposed tools from models devoid of amplification. Also, in the discussion, at lines 359-363, the fact that an orthogonal quantification software is not available is presented as a limitation of the study. Perhaps, having a supplementary reference will show if and when, the study presents limitations.

-CORONATATOR tools has been named multiple times but never used in the comparison. It would be useful to have an explanation for this. Other tools mentioned in the LeTRS paper and available for the identification of TRS sites and junctions (such as SuPER and SARS-CoV-2-leader) were not included in the analysis. Which are the main differences between these tools and those that have been considered? Why were they excluded from the comparison?

Answer: We thank the reviewer for the comment. We decided not to use SuPER and SARS-CoV-2-leader in our study since they were already compared with LeTRS (Xiaofeng Dong X. et al. *Gigascience*. 2022 May 26;11:giac045), showing lower sensitivity than LeTRS. Furthermore, SuPER and SARS-CoV2-leader were mainly developed for RNA-Seq data.

CORONATATOR was not tested as it is more specifically designed for analyzing RNAseq-based metatranscriptomic data. Finally, PERISCOPE has been used as the "gold standard" because it is widely used for SARS-CoV-2 subgenomic analyses, although it is not suitable for output from amplicon ARTIC-based Illumina sequencing approach. The main text was modified to further clarify this aspect (lines 123-124).

Reviewer's answer: while it is now clear the reason to exclude CORONATATOR from this benchmarking, it would be nice to have a clarification on the sentence at lines 122-126: "Thus, in the present study, Periscope (Parker et al. 2021), sgDI-tector (di Gioacchino et al. 2022), LeTRS (Dong et al. 2022) were tested to evaluate their performances in the identification of c- and nc-sgRNA using a dataset obtained with an amplicon ARTIC-based Illumina sequencing approach, in SARS CoV-2 infected Caco-2 cells."

If Periscope is not suitable for output from amplicon ARTIC-based Illumina sequencing approach, why is it used as a "gold standard" in a context in which data have been produced according to this approach? Something similar is also written in the discussion at line 285.

-The provided github directory with the code is not accessible, therefore I could not review the adequacy and correctness of the analyses performed.

Answer: We apologize for the inconvenience, but we are waiting the acceptance and publishing of the article to make the scripts public available.

Reviewer's answer: at the current state, it is not possible to reproduce the bioinformatic analysis only through the methods section, therefore I suggest either to expand it (adding library used, methods applied, ecc.) or to provide a github directory than can help in the reproducibility.

Minor points

Other more targeted suggestions may be the subsequent:

-Please, include citations for the introductory paragraph on SARS-CoV-2 (lines 39-45);

Answer: We addressed this point, citations were added (line 55)

-Explain clearly why sgDI-detector "picks out all the sgRNAs considering that all sgRNAs include the leader sequence", but at the same time, its main principle is not to take into account any TRS sequence (lines 86-87).

Answer: As suggested we modified the description to clearly explain the pipeline (lines 105-109).

Reviewer's answer: Could you please clarify the meaning of this sentence (lines 107-110)? "As matter of fact, the tool detects potential sgRNAs from fragmented reads with insertion, deletion, copy back and hairpin defective viral genome (DVGs) using the DI-tector(Beauclair et al. 2018), and that sgRNA coding for expressed ORFs are much more abundant than DVGs". The last subordinate seems not clear.

-Table 1 might also include a general evaluation of the replicate concordance, including in the same measurement, both c-sgRNAs and nc-sgRNAs fragments.

Answer: We thank the reviewer. The table 1 was replaced with the Fig.2 as suggested, and table 1 was reported as Suppl. Table 1. The number of fragments for c-sgRNAs and nc-sgRNAs was not reported in the Suppl. Table 1 as this table shows the quality of sequencing data.

Reviewer's answer: thank you for the clarification.

-Correction at line 107: it should be LeTRS and sgDI-detector;

Answer: Done (line 468)

At lines 127-128, please explain the source of the total number of fragments chosen for normalization.

Answer: We added a sentence to better explain which source we used (lines 151-152).

Reviewer's answer: thank you for the clarification. I would however suggest using the word "downsampling" instead of "normalization" as the meaning of the second can be misleading in this context.

-Table 2 might be included in the supplementary material while a Figure expressing the percentage quantification for each replicate might substitute it, in order to deliver the message more straightforwardly.

Answer: As previously specified we replaced as suggested with the Fig. 4, and table 2 was reported as Suppl. Table 2A.

Reviewer's answer: Thanks for accepting the suggestion.

-If the concordance of the two replicates at each time point is considered acceptable, from Table 3 onward, both replicates should be shown in the same plot. It might also be useful to explain the reason for having five different timepoints and what we would expect in terms of quantification of ORFs from each of them (see Major points).

Answer: We thank the reviewer for the comment. As suggested, we modified the Figure 5 and we kept the plots for each single replicate in the Suppl. Fig.1A-L and Suppl. Fig. 2A-M. As far as the different timepoints are taken into account, we know very well that from the point of view of the performance of the different tools, little changes. The reason why we decided to show all the timepoints is that we observed differences in expression over time and thought it could be appropriate to show these variations. About quantification, as already mentioned above, our NGS subgenomic analysis was performed by means of an amplicon approach involving several PCR amplifications; we think it is appropriate to keep only the relative comparison between the three bioinformatic tools.

Reviewer's answer: the same thing about the use of the mean (see Major point section) can be applied to Figure 5. Furthermore, it is not clear how the concordance between the replicates has been evaluated. Lines 159-160 state: "In general, the two technical replicates for each time point agreed in the number of identified canonical sgRNAs and in the number of supporting fragments"; but is not supported by a figure or an indication of how this agreement has been evaluated. Perhaps the use of Figure 2B together with a measure of the distance between two replicates may be one of the possible choices. Even the use of Suppl. Table 3A might help in this.

-Include a plot for claim in lines 133-135 possibly with an orthogonal reference next to it.

Answer: We thank the reviewer for the suggestion. We included a new plot (Fig.5 and Suppl. Fig.1AL and Suppl.Fig. 2A-M). As discussed in a previous comment, we prefer not to use an external quantitative reference based on literature data.

-Please clarify, what do you mean in lines 145-146 for common sgRNAs. Does this refer to the independent gRNAs to whom each ORF was assigned?

Answer: We added a sentence to better clarify the concept (line 166)

-It is not straightforward to extrapolate the meaning of Figure 4. It may benefit again from grouping the replicates for each tool and time point (and include single replicate information in the supplementary material). In addition, it would be nice to see as a supplementary figure, for each ORF, a barplot of the quantification across the five time points for the three tools, possibly next to an orthogonal reference (so to better represent claims of lines 158-162).

Answer: As suggested, we modified the Figure 4 now Fig. 5 and we kept the plots for each single replicate in the Suppl. Fig.1A-

L. Moreover, we added in the Suppl.Fig. 2A-M the plot for each ORF across the five time points. As discussed in a previous comment, it was not possible to use an external quantitative reference based on literature data.
Reviewer's answer: refer to the previous point on the use of the mean for two replicates, also valid in this case. The same thing can be extended to Figure 8 and 9.

-Table 4 might benefit from including again, next to nc-sgRNAs quantification, data of c-sgRNAs quantification as percentages so to be able to compare all the information together through the same parameter.

Answer: We thank the reviewer for the comment. The table 4 was replaced as suggested with the Fig.6 and was reported as Suppl. Table 2B.

-The take-home message of Figure 6 is not clear, and the interpretation may not be easy, especially because of low resolution images.

Answer: See comment below.

-Figure 7, as Figure 4, might benefit from the inclusion (as supplementary material) of barplots for each ORF across timepoints. Use a table to represent the quantity of new and previously annotated junctions in each sample at each time point, to simplify the representation.

Answer: As suggested we modified the Figure 6 now Fig. 8 and we kept the plots for each single replicate for each position in the Suppl. Fig. 3A-L and grouped by ORFs in the Suppl. Fig. 4A-L. Moreover, we added in the Suppl. Fig. 5 A-M the plot for each ORF across the five time points. The quantity is reported in the Suppl. Table 3B.

-The discussion is well-articulated, even though, for each annotated junction, a close zoomed figure might help the reader to understand the data supporting the quantification performed by the three tools.

Answer: Thanks to the previous suggestions, we added more punctual figures and the discussion was modified consequently.

Additional minor points

-line 19, there is a typo: "three different softwareS"

-line 39, at this point canonical and non-canonical sgRNAs should be written out in full, as they are not previously mentioned (sentence at line 30 has been deleted)

-line 46, in the subordinate starting with "whereas more differences...", I think something is missing, e.g. "in nc-sgRNA with respect to..."

-lines 72-73, the same citation is repeated multiple times

-lines 96-97, the same citation is repeated multiple times

-lines 157-158, could you please specify in the legend of Figure 3 or in the methods section, how the piechart has been obtained? Is it the average across all the duplicates and all the time points? Please also include percentages (numbers) in the figure itself.

-lines 161-162, one of the named ORFs seem to peak at 24 hpi not at 48 hpi

-lines 207-212, are there orthogonal references at the same infection timepoint that support these nc-sgRNAs or these junctions?

-lines 247-248, the same citation is repeated multiple times

-line 251, the position 27,744-27,779 in ORF 7a/b is cited as previously unknown in literature but at line 232, it is annumerated among the ones confirmed by Lyu et al. Same thing in the discussion respectively at line 342 and 356.

-lines 267-268, Periscope seems not the only one to identify junctions in ORF10 at 96hpi

-line 301, there is a typo: "SARS-VoV-2"

-line 315, figure number is missing.

-line 320, please expand on the temporal trend you were expecting.

-line 325, there are also evidences that show the existence of ORF10 (e.g. Finkel et al.)

-line 328, there is a typo, "the" is repeated twice

-in Figure 8, it could be helpful to mark the positions already confirmed in literature, perhaps with a color legend indicating by which paper they have been confirmed.

Reviewer #2 (Comments to the Authors (Required)):

I have no further questions. I am looking forward to see the publication.

Reviewer #1 (Comments to the Authors (Required)):

The work presented by Lavezzari et al. has improved from the first revision, however there are still some points that may benefit from clarifications and expansions. Finally, I would suggest a professional proof reading.

Major points

-Figures that are mainly made up of tables of fragment counts and percentages are not delivering a straightforward and quantitative message. I would like to suggest an easier data visualization approach, by using plots (e.g. concordance plots or barplots) such as in the case of Table 1, 2 and 4. These can be supplementary data, leaving the burden of delivering the message to specific plots that best address the scope.

Answer: We thank the reviewer for the comment. The tables 1, 2 and 4 were replaced as suggested with the Figures 2, 4 and 6 respectively. Tables were reported as Suppl. Table 1, Suppl. Table 2A and 2B respectively.

Reviewer's answer: the inclusion of the new figures delivers a much clearer message to the reader, thank you for implementing them. Furthermore, in every plot where data is represented by a mean of two duplicates, the authors should show, beside the mean, also the points representing the value of each replicate (e.g. <https://doi.org/10.1038/s41551-017-0079>). A mean of two points is qualitatively delivering an information but it is not considerable statistically robust.

Answer: Thanks a lot for the suggestion, Figure 4 and 6 were modified according to the suggestion. Figure 2 already represents the single replicates values.

-It would be helpful to provide a quantification control for the expression of different ORFs. In most of the plots, the expression results from the considered tools are compared by themselves but not to an orthogonal quantitative reference. Performing further experiments is not necessary, as data for sgRNAs quantification are already available in the form of Northern Blots (e.g. Ogando et al., doi: 10.1099/jgv.0.001453) or Nanopore Sequencing databases.

Answer: We carefully read Ogando et al. paper but we are not sure that their transcripts relative abundance can be applied as orthogonal quantitative reference in our study. In fact, they used a different in vitro model and different time points (their last time point is 24hpi while in our study we considered a longer interval until 96hpi). Moreover, we would like to highlight that our NGS subgenomic analysis was performed through an amplicon approach that implies several PCR amplifications. For this reason, we believe that even if the genomic data had been available the correlation among the bioinformatic quantitation with the viral load calculated using a quantitative reference could be misrepresentative, that's why we would prefer to maintain a relative comparison between the three bioinformatics tools.

Reviewer's answer: Given that the PCR creates itself an artefact on the quantification, it is exactly for this reason that I would like to see an orthogonal method (a single timepoint at 24 hpi is sufficient, and other authors have quantified sgRNAs in CaCo2, as <https://doi.org/10.1016/j.molcel.2021.02.036>). Indeed, this will suggest how close or how far is the quantification of the three proposed tools from models devoid of amplification. Also, in the discussion, at lines 359-363, the fact that an orthogonal quantification software is not available is presented as a limitation of the study. Perhaps, having a supplementary reference will show if and when, the study presents limitations.

Answer: We thank the Reviewer for the comment. We have carefully read the suggested publication (including that by the first round of revision), but the authors used total RNA and RNA sequencing methods and not the amplicon approach as in our work. Moreover, the northern blot was used only to verify the presence of sgRNA and not to directly quantify the sgRNA (no OD/quantification data were reported for northern blot analysis). In supplementary figures they simulated a “Northern image according to the sequence lengths of Nanopore long-reads” but again, they did not use these simulated data. As far as we understood, also in Wang et al. the quantification was made counting and comparing the reads percentages detected by both Nanopore and Illumina RNAseq data, without referring to any northern blotting data. Thus, it seems that our analysis approach is similar to that reported. For this reason, we prefer to stick to our experimental data without using any external/literature reported quantification reference.

-CORONATATOR tools has been named multiple times but never used in the comparison. It would be useful to have an explanation for this. Other tools mentioned in the LeTRS paper and available for the identification of TRS sites and junctions (such as SuPER and SARS-CoV-2-leader) were not included in the analysis. Which are the main differences between these tools and those that have been considered? Why were they excluded from the comparison?

Answer: We thank the reviewer for the comment. We decided not to use SuPER and SARS-CoV-2-leader in our study since they were already compared with LeTRS (Xiaofeng Dong X. et al. Gigascience. 2022 May 26;11:giac045), showing lower sensitivity than LeTRS. Furthermore, SuPER and SARS-CoV2-leader were mainly developed for RNA-Seq data. CORONATATOR was not tested as it is more specifically designed for analyzing RNAseq-based metatranscriptomic data. Finally, PERISCOPE has been used as the "gold standard" because it is widely used for SARS-CoV-2 subgenomic analyses, although it is not suitable for output from amplicon ARTIC-based Illumina sequencing approach. The main text was modified to further clarify this aspect (lines 123-124).

Reviewer's answer: while it is now clear the reason to exclude CORONATATOR from this benchmarking, it would be nice to have a clarification on the sentence at lines 122-126: "Thus, in the present study, Periscope (Parker et al. 2021), sgDI-tector (di Gioacchino et al. 2022), LeTRS (Dong et al. 2022) were tested to evaluate their performances in the identification of c- and nc-sgRNA using a dataset obtained with an amplicon ARTIC-based Illumina sequencing approach, in SARS CoV-2 infected Caco-2 cells."

If Periscope is not suitable for output from amplicon ARTIC-based Illumina sequencing approach, why is it used as a "gold standard" in a context in which data have been produced according to this approach? Something similar is also written in the discussion at line 285.

Answer: Thanks for the comment. After contacting Periscope developers for asking some specification and implementation for making it more suitable for Illumina amplicon analysis, we decided to use it for the analysis, as in the newest implemented version of the software it also works on amplicon ARTIC-based Illumina sequencing approach (as written in their article Parker et al. 2021). Moreover, as at the time it was the most cited and used software for the research of sub-genomics in the different kind of sequencing data (mainly Nanopore) we decided to use it as “gold standard”. In the discussion we argue that it is not suitable for output from amplicon ARTIC-based Illumina sequencing approach as its main purpose is the analysis of Nanopore data.

-The provided github directory with the code is not accessible, therefore I could not review the adequacy and correctness of the analyses performed.

Answer: We apologize for the inconvenience, but we are waiting the acceptance and publishing of the article to make the scripts public available.

Reviewer's answer: at the current state, it is not possible to reproduce the bioinformatic analysis only through the methods section, therefore I suggest either to expand it (adding library used, methods applied, ecc.) or to provide a github directory than can help in the reproducibility.

Answer: We provide a zip folder to the Reviewer (and not for publication) with the script deposited on GitHub.

Minor points

Other more targeted suggestions may be the subsequent:

-Please, include citations for the introductory paragraph on SARS-CoV-2 (lines 39-45);

Answer: We addressed this point, citations were added (line 55)

-Explain clearly why sgDI-detector "picks out all the sgRNAs considering that all sgRNAs include the leader sequence", but at the same time, its main principle is not to take into account any TRS sequence (lines 86-87).

Answer: As suggested we modified the description to clearly explain the pipeline (lines 105-109).

Reviewer's answer: Could you please clarify the meaning of this sentence (lines 107-110)? "As matter of fact, the tool detects potential sgRNAs from fragmented reads with insertion, deletion, copy back and hairpinning defective viral genome (DVGs) using the DI-tector (Beauclair et al. 2018), and that sgRNA coding for expressed ORFs are much more abundant than DVGs". The last subordinate seems not clear.

Answer: Apologies to the Reviewer but there has been a typo from the last revision. We have now changed the sentence (lines 96-99).

-Table 1 might also include a general evaluation of the replicate concordance, including in the same measurement, both c-sgRNAs and nc-sgRNAs fragments.

Answer: We thank the reviewer. The table 1 was replaced with the Fig.2 as suggested, and table 1 was reported as Suppl. Table 1. The number of fragments for c-sgRNAs and nc-sgRNAs was not reported in the Suppl. Table 1 as this table shows the quality of sequencing data.

Reviewer's answer: thank you for the clarification.

-Correction at line 107: it should be LeTRS and sgDI-detector;

Answer: Done (line 468)

At lines 127-128, please explain the source of the total number of fragments chosen for normalization.

Answer: We added a sentence to better explain which source we used (lines 151-152).

Reviewer's answer: thank you for the clarification. I would however suggest using the word "downsampling" instead of "normalization" as the meaning of the second can be misleading in this context.

Answer: Thanks for the suggestion, we modified the word (line 137).

-Table 2 might be included in the supplementary material while a Figure expressing the percentage quantification for each replicate might substitute it, in order to deliver the message more straightforwardly.

Answer: As previously specified we replaced as suggested with the Fig. 4, and table 2 was reported as Suppl. Table 2A.

Reviewer's answer: Thanks for accepting the suggestion.

-If the concordance of the two replicates at each time point is considered acceptable, from Table 3 onward, both replicates should be shown in the same plot. It might also be useful to explain the reason for having five different timepoints and what we would expect in terms of quantification of ORFs from each of them (see Major points).

Answer: We thank the reviewer for the comment. As suggested, we modified the Figure 5 and we kept the plots for each single replicate in the Suppl. Fig.1A-L and Suppl. Fig. 2A-M. As far as the different timepoints are taken into account, we know very well that from the point of view of the performance of the different tools, little changes. The reason why we decided to show all the timepoints is that we observed differences in expression over time and thought it could be appropriate to show these variations. About quantification, as already mentioned above, our NGS subgenomic analysis was performed by means of an amplicon approach involving several PCR amplifications; we think it is appropriate to keep only the relative comparison between the three bioinformatic tools.

Reviewer's answer: the same thing about the use of the mean (see Major point section) can be applied to Figure 5. Furthermore, it is not clear how the concordance between the replicates has been evaluated. Lines 159-160 state: "In general, the two technical replicates for each time point agreed in the number of identified canonical sgRNAs and in the number of supporting fragments"; but is not supported by a figure or an indication of how this agreement has been evaluated. Perhaps the use of Figure 2B together with a measure of the distance between two replicates may be one of the possible choices. Even the use of Suppl. Table 3A might help in this.

Answer: Thanks for the comment. Figure 5 was modified as suggested before. Concordance among replicates have been evaluated calculating the Intraclass Correlation Coefficient (ICC). We added the values of the aforementioned analysis in Supplementary material section, Table S2A and we added a clarification sentence in the main text (lines 143-145, line 420).

-Include a plot for claim in lines 133-135 possibly with an orthogonal reference next to it.

Answer: We thank the reviewer for the suggestion. We included a new plot (Fig.5 and Suppl. Fig.1AL and Suppl.Fig. 2A-M). As discussed in a previous comment, we prefer not to use an external quantitative reference based on literature data.

-Please clarify, what do you mean in lines 145-146 for common sgRNAs. Does this refer to the independent gRNAs to whom each ORF was assigned?

Answer: We added a sentence to better clarify the concept (line 166)

-It is not straightforward to extrapolate the meaning of Figure 4. It may benefit again from grouping the replicates for each tool and time point (and include single replicate information in the supplementary material). In addition, it would be nice to see as a supplementary figure, for each ORF, a barplot of the quantification across the five time points for the three tools, possibly next to an orthogonal reference (so to better represent claims of lines 158-162).

Answer: As suggested, we modified the Figure 4 now Fig. 5 and we kept the plots for each single replicate in the Suppl. Fig. 1A-L. Moreover, we added in the Suppl. Fig. 2A-M the plot for each ORF across the five time points. As discussed in a previous comment, it was not possible to use an external quantitative reference based on literature data.

Reviewer's answer: refer to the previous point on the use of the mean for two replicates, also valid in this case. The same thing can be extended to Figure 8 and 9.

Answer: Thanks a lot for the suggestion, we modified the Figures 5, 8 and 9 as suggested with bars representing the mean and points representing the single replicates.

-Table 4 might benefit from including again, next to nc-sgRNAs quantification, data of c-sgRNAs quantification as percentages so to be able to compare all the information together through the same parameter.

Answer: We thank the reviewer for the comment. The table 4 was replaced as suggested with the Fig.6 and was reported as Suppl. Table 2B.

-The take-home message of Figure 6 is not clear, and the interpretation may not be easy, especially because of low resolution images.

Answer: See comment below.

-Figure 7, as Figure 4, might benefit from the inclusion (as supplementary material) of barplots for each ORF across timepoints. Use a table to represent the quantity of new and previously annotated junctions in each sample at each time point, to simplify the representation.

Answer: As suggested we modified the Figure 6 now Fig. 8 and we kept the plots for each single replicate for each position in the Suppl. Fig. 3A-L and grouped by ORFs in the Suppl. Fig. 4A-L. Moreover, we added in the Suppl. Fig. 5 A-M the plot for each ORF across the five time points. The quantity is reported in the Suppl. Table 3B.

-The discussion is well-articulated, even though, for each annotated junction, a close zoomed figure might help the reader to understand the data supporting the quantification performed by the three tools.

Answer: Thanks to the previous suggestions, we added more punctual figures and the discussion was modified consequently.

Additional minor points

-line 19, there is a typo: "three different softwares"

Answer: We checked and the noun "software" has been written as uncountable in line 19 as well as in all the text.

-line 39, at this point canonical and non-canonical sgRNAs should be written out in full, as they are not previously mentioned (sentence at line 30 has been deleted)

Answer: Now the full name has been added (lines 35-36).

-line 46, in the subordinate starting with "whereas more differences...", I think something is missing, e.g. "in nc-sgRNA with respect to..."

Answer: The phrase is now modified (line 39).

-lines 72-73, the same citation is repeated multiple times

Answer: Thanks a lot, we removed them (lines 62-63).

-lines 96-97, the same citation is repeated multiple times

Answer: Thanks a lot, we removed them (line 86).

-lines 157-158, could you please specify in the legend of Figure 3 or in the methods section, how the piechart has been obtained? Is it the average across all the duplicates and all the time points? Please also include percentages (numbers) in the figure itself.

Answer: We thank the reviewer for the suggestion. We clarify in the figure legend that the piechart shows the average of each time point and we added the percentages (line 446).

-lines 161-162, one of the named ORFs seem to peak at 24 hpi not at 48 hpi

Answer: We thank the Reviewer, but the text is correct. However, we noticed an inaccuracy in the figure S2 that has now uploaded in the correct version.

-lines 207-212, are there orthogonal references at the same infection timepoint that support these nc-sgRNAs or these junctions?

Answer: See major point answer to the comment.

-lines 247-248, the same citation is repeated multiple times

Answer: Thanks a lot, we removed them (lines 218-219).

-line 251, the position 27,744-27,779 in ORF 7a/b is cited as previously unknown in literature but at line 232, it is annumerated among the ones confirmed by Lyu et al. Same thing in the discussion respectively at line 342 and 356.

Answer: Thanks a lot for the observation, there was an error in reporting the positions during the revisions (line 222).

-lines 267-268, Periscope seems not the only one to identify junctions in ORF10 at 96hpi

Answer: Thanks a lot for the observation, we modified the sentence (line 235).

-line 301, there is a typo: "SARS-VoV-2"

Answer: Modified (line 265).

-line 315, figure number is missing.

Answer: Thanks a lot, we added it (line 279).

-line 320, please expand on the temporal trend you were expecting.

Answer: Modified (line 284).

-line 325, there are also evidences that show the existence of ORF10 (e.g. Finkel et al.)

Answer: We thank the reviewer for the observation. The existence of ORF10 is highly debated. We added a sentence reporting the suggested reference indicating the existence of ORF10 (Line 291).

-line 328, there is a typo, "the" is repeated twice

Answer: Modified (line 292).

-in Figure 8, it could be helpful to mark the positions already confirmed in literature, perhaps with a color legend indicating by which paper they have been confirmed.

Answer: We thank the Reviewer for the suggestion, we modified the figure according to. We marked by a red square the positions already identified in literature. We added the red square with the same function also in Figure S3.

Reviewer #2 (Comments to the Authors (Required)):

I have no further questions. I am looking forward to see the publication.

Answer: We thank the Reviewer.

August 29, 2023

RE: Life Science Alliance Manuscript #LSA-2023-02017-TRR

Dr. Elena Pomari
IRCCS Ospedale Sacro Cuore Don Calabria
Viale L. Rizzardi 4
Negrar di Valpolicella 37024
Italy

Dear Dr. Pomari,

Thank you for submitting your revised manuscript entitled "Comparative analysis of bioinformatics tools to characterize SARS-CoV-2 subgenomic RNAs". We would be happy to publish your paper in Life Science Alliance pending final revisions necessary to meet our formatting guidelines.

- please add your main, supplementary figure, and table legends to the main manuscript text after the references section
- please add the Twitter handle of your host institute/organization as well as your own or/and one of the authors in our system
- please be sure to mention all panels in figures in their legends
- please correct callouts in the manuscript text to match panels in figures
- we encourage you to revise the figure legends for figures 2, 5, 7, 8, and 9 such that the figure panels are introduced in an alphabetical order
- please add an Author Contributions section to your main manuscript text
- please add a conflict of interest statement to your main manuscript text
- the github link should be made publicly accessible at this point

Figure checks:

- panels in your supplementary figures and Fig. 7 are not in alphabetical order. Please correct

A. FINAL FILES:

B. MANUSCRIPT ORGANIZATION AND FORMATTING:

Sincerely,

September 12, 2023

RE: Life Science Alliance Manuscript #LSA-2023-02017-TRRR

Dr. Elena Pomari
IRCCS Ospedale Sacro Cuore Don Calabria
Viale L. Rizzardi 4
Negrar di Valpolicella 37024
Italy

Dear Dr. Pomari,

Thank you for submitting your Research Article entitled "Comparative analysis of bioinformatics tools to characterize SARS-CoV-2 subgenomic RNAs". It is a pleasure to let you know that your manuscript is now accepted for publication in Life Science Alliance. Congratulations on this interesting work.

DISTRIBUTION OF MATERIALS:

Again, congratulations on a very nice paper. I hope you found the review process to be constructive and are pleased with how the manuscript was handled editorially. We look forward to future exciting submissions from your lab.

Sincerely,
